# On the replicability of diffusion weighted MRI-based brain-behavior models
Raviteja Kotikalapudi [1,2,3] ✉, Balint Kincses [1,2], Giuseppe Gallitto [1,2], Robert Englert[2,4], Kevin Hoffschlag[1,2], Jialin Li [1,2,5], Christian Büchel [6], Ulrike Bingel [1,2] & Tamas Spisak [2,4] ✉

Replicability of anatomical and functional MRI-based inter-individual BWAS has been extensively discussed recently. This study reports a comprehensive evaluation of BWAS replicability based on structural connectomes (streamlines, FA, MD, RD, AD). Overall, 36%(21/58) of brain-phenotype associations were replicable in the HCP dataset with atleast one of the DWI metrics, and a discovery sample size n ≤ 425 (total sample size = discovery + replication samples). Temporally stable, trait-like phenotypes were found to be more replicable (50%, 16/32), than state-like measures (19%, 5/26). Streamline-based connectomes (SC) provided the highest replicability across all metrics (29% and 42% of phenotypes in the HCP and AOMIC datasets, respectively). In line with theoretical expectations, replicability was found to be directly related to effect size. Phenotypes that needed n > 400 discovery samples to replicate displayed very low effect sizes <2% variance). Effect size magnitudes replicated well with associations that explained more than ~5% variance, typically requiring a discovery n < 300. Our results suggest that trait-like phenotypes can show good replicability with moderate sample sizes and warrant that models that require n > 425 samples will necessarily display limited practical relevance due to their small predictive performance. Large sample sizes remain crucial for explainability and for assessing fairness and generalizability to new populations.

There is a growing interest in establishing links between individual differences in phenotypic traits and measures of brain structure and function through brain-wide association studies (BWAS). Despite their potential, BWAS have been shown to face challenges related to replicability and, in many cases, require large samples to be reproducible[1]. In contrast to mass-univariate BWAS, multivariate statistical learning techniques can achieve much higher effect sizes and can therefore be replicable with low to moderate sample sizes in several cases[2–4]. This has recently been demonstrated with anatomical and functional MRI data such as cortical thickness, and functional connectivity [2]. However, a notable gap exists in our knowledge about the replicability of multivariate BWAS based on diffusion-weighted (DWI) MRI data. As DWI remains the primary non-invasive technique to offer insights into the microstructural integrity of neural pathways, replicability benchmarks for various DWI-based brain features are urgently needed to be able to draw informed decisions when planning, funding and reviewing future studies.

Here we report the results of a systematic, large-scale, theoretical and empirical replicability analysis[2] of DWI-based multivariate BWAS. Our work encompasses machine learning analyses on streamline-based structural connectivity (SC), as well as microstructural connectivity measures derived from fractional anisotropy (FA), radial (RD) and axial (AD) diffusivity, and apparent diffusion coefficient (ADC), in case of 58 behavioral and psychometric phenotypes from the Human Connectome Project (HCP)[5,6] and 19 behaviouaral measures from the Amsterdam Open MRI Collection (AOMIC)[7], involving millions of resampling-based model fits and thousands of core-hours of pre-processing and model trainings. We contrast empirical effect sizes and replicability estimates to theoretical predictions and discuss the key requirements for establishing replicable DWI-phenotype associations.

## Results and discussion

Using the HCP, we tested the replicability of 5 different DWI connectome-based brain-behavior models (for list of behavior measures

[1]Department of Neurology, University Medicine Essen, Essen, Germany. [2]Center for Translational Neuro- and Behavioral Sciences (C-TNBS), University Medicine Essen, Essen, Germany. [3]Department of Neurology, University Medicine Goettingen, Goettingen, Germany. [4]Department of Diagnostic and Interventional Radiology and Neuroradiology, University Medicine Essen, Essen, Germany. [5]Max Planck School of Cognition, Leipzig, Germany. [6]University Medical Center Hamburg, Hamburg, Germany. ✉e-mail: raviteja.kotikalapudi@uk-essen.de; tamas.spisak@uk-essen.de

and their categorization into trait-like and state-like measures, see Supplementary Table 1), namely, streamline count-based (SC) and FA, radial diffusivity (RD), axial diffusivity (AD) and ADC-based SC measures. Our analyses focused on estimating the minimum discovery sample size needed to establish multivariate DWI-based brain-phenotype associations that are replicable in an independent sample (replications sample) that is of the same size as the discovery sample). For our study, we adapted the methodology of Spisak et al.[2] and used the replication probability threshold of $P_{replication} > 0.8$, meaning that the identified brain-phenotype association has a probability of greater than 80% to be significant ($p < 0.05$) in the replication study, given that it was significant in the discovery dataset. For a given sample size and phenotype, we estimated $P_{replication}$ by repeatedly sampling non-overlapping, equally sized discovery and replication sets and testing the significance of the established associations in both. In the discovery phase, we fitted a Ridge regression model (for a supplementary analysis benchmarking other approaches, see Supplementary Table 2). Optimal model regularization, as well as discovery effect sizes, were estimated in a nested cross-validation framework, to avoid biased estimates. The pipeline for the replication analysis is shown in Supplementary Fig. 1. In addition to $P_{replicability}$, we also investigated how well the magnitude of the effect sizes replicates; an approach independent of arbitrary significance thresholds. We performed these analyses for 58 different variables from the HCP with variable discovery sample sizes ($n = 25$ to $425$ in steps of $25$). Finally, we also provide a theoretical sample size estimate given the empirical effect sizes of phenotypes. To investigate how well our results generalize to other datasets, we repeated all analyses for the best performing DWI-metric (SC, based on average sample sizes required for replication) in another dataset—AOMIC with maximum sample split size of $n = 425$ (discovery + replication = 900), and 19 phenotypes.

## Performance of DWI-connectomes with HCP dataset

The percentage of brain-phenotypes associations replicable with a sample size of $n = 425$ (both for the discovery and replication set) were comparable across DWI metrics (Fig. 1A–D, Supplementary Fig. 2, Supplementary Table 3–6), on average 28%. When considering successful replication as being replicable in case of at least one DWI metric, 36% (21/58) of phenotypes were replicable. As the inherently slow rate of plasticity in white matter microstructure—as captured by DWI cannot reflect state-like phenotypes with rapid fluctuations (e.g., emotional states), we differentiated between enduring and relatively stable phenotypes, which represent long-lasting traits, and transient/fluctuating, short-lived state-like characteristics, which constitute temporary states[8]. When considering state-like phenotypes, we found that 50% of them were replicable with $n \leq 425$, with at least one DWI metric. In the state-like category, 19% of phenotypes were found to be replicable with the same criteria.

When comparing different DWI-metrics, SC-based models were the most economic in terms of discovery sample sizes needed to produce significant predictions in both the discovery and the replication samples. Across all phenotypes that were replicable with a discovery set of $n \leq 425$, SC-models required $n = 171$ samples on average (for trait-like phenotypes $n = 150$ and for state-like $n = 325$). The empirical sample size requirments and effect-size driven theoretical sample size requires were also observed in good correspondence with each other ($r > 0.90$, Table 1).

The effect sizes for the discovery and replication datasets across all phenotypes and all sample sizes are presented in Supplementary Data 1–5. In general, the magnitude of effect sizes across sample sizes (measured as Pearson's correlation between prediction and obsrevation—per phenotype—resulting in 58 $r$ values) obtained during the discovery phase were highly correlated to those seen in the replication phase (mean $r$ across phenotypes = 0.70–0.74 across DWI connectome types). With smaller sample sizes (e.g., $n < 250$) and smaller effect sizes (e.g., $r < 0.2$), the discovery effect tended to be smaller than the replication effect size (range of difference, $\Delta r = -0.27$ to $-0.06$, Fig. 1E, Supplementary Fig. 3); likely a consequence of different sample characteristics. At higher sample sizes

($n > 250$), the discovery and replication effect sizes differed much less ($\Delta r = -0.05$ to $-0.04$), as the larger and more representative discovery sample allowed better generalization to the replication sample (Note that the discovery and replication sets were randomly sampled without matching any variables).

## Streamline connectivity (SC) validity with AOMIC dataset

We investigated the replicability of a total of 19 additional phenotype in the AOMIC dataset, focusing on streamline connectivity (SC, for all phenotypes see Fig. 1G). Overall, 42% (8/19) of the phenotypes were replicable in this dataset with $n \leq 425$, with an average sample size requirement of $n = 228$ (please see Supplementary Table 7 for more details and see Supplementary Table 8 *for other metrics, e.g., FA*). The consistency of our SC findings was notably reinforced through a comparative analysis with the AOMIC dataset, which served as a secondary validation cohort. Key observations from the primary HCP dataset were largely mirrored in the AOMIC data; for instance, the average sample sizes derived from the HCP dataset were comparable to those in the AOMIC dataset, suggesting a robust pattern across different data sources. Furthermore, the predictive performance of the SC model, initially evaluated using HCP data, demonstrated analogous efficacy when applied to the AOMIC dataset. This cross-dataset congruence extended to trait replicability, where the patterns of trait replication observed in the HCP models were also evident in the AOMIC dataset. Collectively, these parallel outcomes across both primary (HCP) and secondary (AOMIC) datasets provide strong evidence for the generalizability and reliability of our initial results.

## Exploratory analysis for improved performance using a higher-resolution atlas

Our primary replicability analyses on the HCP dataset utilized the 84-region Desikan Killiany brain atlas[9], which is the default option of the software FreeSurfer. To investigate how the choice of brain atlas impacts our results, we performed exploratory analyses on the AOMIC dataset using a higher-resolution 162-node atlas (the Destrieux atlas[10]) indicated a potential for improved replicability (increasing from 42% to 47% of phenotypes replicated) and the prediction of one more additional trait namely "intelligence structured test for memory" (please see Supplementary Table 9). This also resulted in an improvement of mean sample sizes requirements for replicability, from $n = 228$ to $n = 217$ on avergae (mean across all replicable phenotypes), with ~5% less samples required. Although the generalizability of this improved performance requires further validation across diverse datasets, these findings indicate that employing higher-resolution parcellations could offer valuable information gains for future replicability studies.

## Additional confounder testing on the Human Connectome Project

While the current study's primary focus is replicability, there are several other important requirements that a DWI-based biomarker candidate must fulfill. In an additional analysis, detailed in Supplementary Tables 10, 11, we perfomed an initial evaluation of one of these important requirements, confounding bias. Specifically, we investigated how much our predictions within the HCP dataset were biased by total intracranial volume (TIV) with a novel statistical test for detecting confounding bias in multivariate predictive models (Python package "mlconfond")[11]. While our results indicated that none of the phenotypes were fully driven by TIV, most of them were found to be partially biased towards TIV to some degree. While these results are initial in their nature, they highlight that the appropriate handling of confounders—including, but not limited to, TIV—are an essential next steps towards building DWI-based predictive models with true translational potential.

## Interpretation of brain-phenotype characterization

To interpret the contribution of individual features to model predictions, we performed SHapley Additive exPlanations (SHAP)[12] analysis on models trained on the full HCP dataset ($n = 900$) for phenotypes demonstrating replicable prediction performance with the SC DWI connectome. Features

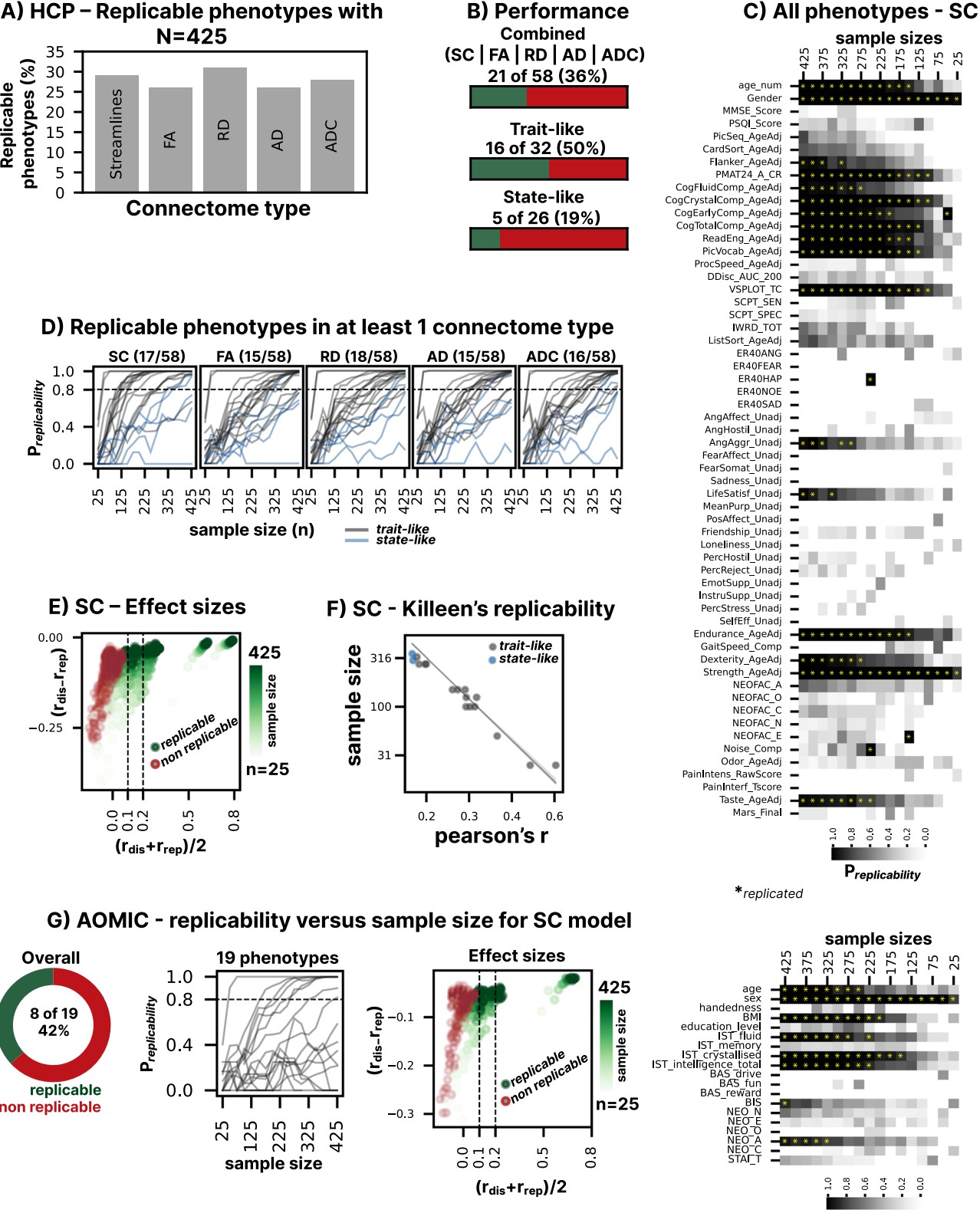

**A) HCP – Replicable phenotypes with N=425**

**B) Performance**

**C) All phenotypes - SC**

**D) Replicable phenotypes in at least 1 connectome type**

**E) SC – Effect sizes**

**F) SC - Killeen's replicability**

**G) AOMIC - replicability versus sample size for SC model**

were ranked by their mean absolute SHAP value to quantify overall importance. For each model, we identified the top 10 most influential white matter tract features. We found that these top-ranking features exhibited a significant correlation with their target behavioral measures across the majority of replicable phenotypes ($p < 0.05$), with the life satisfaction model being a notable exception. The complete list of top 10 features for each

model, including their mean absolute SHAP values and feature-target correlation statistics, is provided in Supplementary Fig. 4.

Our results, show that sample size (i.e., number of samples needed to finalize the predictive model = discovery samples, total samples = discovery + replication samples) requirements for replicable DWI-based BWAS are generally better than those observed with anatomical MRI data

**Fig. 1 | Replicability of DWI-predictive models of behavior with small-to-medium sample sizes. A** Percentage of replicable brain-phenotype models within the HCP dataset. $N = 425$ refers to the maximum number of samples used to train-test predictive models in the discovery phase. The discovery set and replication set were always split equally, that also means, total samples = discovery + replication sets. Overall performance across DWI metrics is comparable. **B** Overall, 21 out of 58 phenotype models were replicable (with $n \leq 425$), including both trait-like and state-like measures. Overall, here refers to considering a brain-behavior model replicable, provided it was replicable with at least one of the 5 connectome types (SC|FA|RD|AD|ADC). Brain-based characterizations of long-term traits were more replicable in comparison to characterizations of the short-term, fluctuating nature of state-like measures. **C** Heatmap displaying $p_{replicability}$ all the HCP behavioral measures and at what sample sizes they have demonstrated replicability. **D** A closer look at the 21 phenotypes that were replicable by atleast one connectome type. In connectome types where a particular phenotype of the 21 replicable phenotypes is not replicable, the line does not cross 0.8 mark (y-axis). It can be observed that SC models saturate

faster in comparison with their connectome counterparts. **E** Effect sizes in the SC models. The plot shows the relationship between average effect sizes (i.e., average between effect sizes achieved in the discovery and replication datasets) versus delta (i.e., discovery effect sizes—replication effect sizes). The plot demonstrates that as the mean effect sizes increases, the delta decreases, especially for the phenotypes that were replicable under 425 samples (blue dots). For the rest, it can also be observed that for lower sample sizes, the delta is larger (for average $r < 0.1$). **F** Correspondence of empirical (points) and theoretical (line) sample size requirements, plotted against the effect size in the replication set. As the plot shows, brain-phenotype associations that replicate at higher sample sizes tend to display lower effect sizes. Associations that only replicate with $n > 425$ will likely explain only around 2% of variance in the phenotype and, thus, have low practical relevance. **G** Replication analysis for the SC models based on the AOMIC dataset. Similar to the HCP dataset, DWI-based connectivity showed comparable brain-phenotype replication models, in terms of sample sizes as well as effect sizes.

but worse than in case of resting state functional connectivity. For instance, to train a replicable model for the NIH total cognitive ability score $n = 75$, 125, and 200 samples are needed with functional (resting state), DWI and anatomical (cortical thickness) data, respectively (data for anatomical and functional data taken from ref. [2]). Similar to reports about functional and anatomical measures[1,2,13], we observed that a relatively high proportion of the investigated brain-phenotype models were either not significant or not replicable with the maximum investigated sample size ($n = 425$, both in the discovery and replication set). Importantly, our theoretical analysis suggests that increasing sample size for these phenotypes to reach replicability is not likely to improve on the usefulness and practical relevance of these associations, as the effect sizes for such measures will likely be low (below 2% of variance explained). This sobering result warrants that the choice of the appropriate target phenotype is a crucial requirement for replicable BWAS. The question of 'what is an appropriate target phenotype' can be discussed from multiple aspects.

First, it is important to consider whether the phenotype of interest is relevant at all to the investigated brain measure. DWI-based brain features primarily capture microstructural characteristics of brain tissue, that are only subject to slow changes (due to development or plasticity) and are less prone to rapid alterations or fluctuations over time. Thus, we can expect that DWI data can only predict those measures reliably that are also persistent in time. From this perspective, phenotypes related to cognition, motor skills and demographics (e.g., age and gender) are well-suited to be predicted from DWI due to their inherent stability and/or trait-like nature. Indeed, we observed that a large number of phenotypes belonging to such categories (labeled as cognition, memory, motor, sensory and personality traits) were replicable when trained with $n \leq 425$. In contrast, phenotypic measures related to emotions (account for ~45% of all phenotypes in HCP1200 dataset and no state-like in AOMIC dataset) may exhibit relatively rapid changes over time[8]. The inherent static information representation via tractography might not be sufficient for explaining greater effect sizes in such phenotypes.

Second, contrasting empirical and theoretical replicability estimates (calculated with Kileen's replication probability) substantiated that the replicability of multivariate DWI-based brain-behavior models are primarily a function of their effect sizes. The strong dependence of replicability on effect size implies that, as substantiated by Gratton et al.[14] and others[2,15], there are "two paths towards reliability". Next to increasing sample sizes (possibly to the thousands), the reliability of the target phenotype, and its relevance to the brain measure, is also of central importance to improve the replicability. It is therefore crucial to acknowledge that when replicability is achieved primarily through the path of very large sample sizes, this often corresponds to smaller underlying effect sizes attributable to white matter microstructural properties alone. Consequently, the practical significance and real-world utility of such findings necessitate careful evaluation, even if statistical replicability is met. Phenotypes that are not only highly relevant to the brain feature of interest (e.g., trait measures for DWI) but also display a high test-retest reliability can be expected to display the highest effect sizes

and will be replicable with small sample sizes. This has been exemplified by several previously reported multivariate neural signatures that showed strong, replicable effect sizes with various behavioral variables, even though they were trained on small samples, sometimes fewer than 50 participants[3,16]. We argue that such models are the most promising candidates for individual-level predictions guiding clinical care and personalized medical approaches and our findings hint that the effect sizes and replicability necessary for these applications is not impossible to achieve, at least for the most reliable and stable phenotypes. This of course does not imply that only trait-like phenotypes are suitable for predictive modeling in general. However, for more dynamic, state-like phenotypes (like emotional state), approaches leveraging neural processes with faster dynamics, like fMRI, EEG, and MEG, are more appropriate than DWI, which is inherently limited by the slow rate of change/plasticity in white matter microstructure. For example, while the general tendency towards anxiety is a trait-like characteristic, the intensity of anxiety experienced in response to a specific social situation is a state-like attribute that fMRI task-based brain-derived phenotypes could explain more reliably that structural models. Given that mental health often involves a combination of trait-like predispositions and state-like fluctuations, it is important to note that achieving a more comprehensive understanding, diagnosis, and treatment in this domain often requires predictive performance based on both types of variables.

While streamline count (SC), or streamline-based connectivity, consistently yielded more replicable phenotypes compared to other connectome metrics (FA, RD, AD, and ADC) across both datasets, we acknowledge that these findings might be specific to the processing pipeline used in this work. Since SC is heavily dependent on analysis parameters such as the tractography algorithm, the number of seeds, and the total number of streamlines, our results should be interpreted with caution. Furthermore, we must consider that the sample sizes required for replication could also be influenced by the presence of confounders. Therefore, future studies should not only continue to explore the predictive power of other diffusion metrics —as they are crucial for a comprehensive understanding of brain mechanisms—but also investigate the specific effects of these confounders on the required sample sizes for replicable findings.

Besides replicability, there are many other substantial challenges ahead towards robust and clinically useful DWI-based models, including between-site differences[17,18], dataset shift[19], generalizability and fairness across contexts and subpopulations, as well as neuroscientific validity and interpretability. To solve these challenges will likely require high precision target measures, innovatative new modeling and feature processing methods, and, even for highly replicable models, large, representative samples.

## Methods
### Magnetic resonance imaging protocol
Detailed information regarding the T1-weighted and DWI sequences are mentioned elsewhere (https://www.humanconnectome.org/, HCP_S1 200_Release_Reference_Manual.pdf). In brief, T1-weighted scans were

**Table 1 | Evaluation of empirical replicability with theoretical effect-based sample replicability**

| PHENOTYPES | STREAMLINES | | | | FA | | | | RD | | | | AD | | | | ADC | | | |
|---|---|---|---|---|---|---|---|---|---|---|---|---|---|---|---|---|---|---|---|---|
| | $r_{dis}$ | $r_{rep}$ | $n_E$ | $n_T$ | $r_{dis}$ | $r_{rep}$ | $n_E$ | $n_T$ | $r_{dis}$ | $r_{rep}$ | $n_E$ | $n_T$ | $r_{dis}$ | $r_{rep}$ | $n_E$ | $n_T$ | $r_{dis}$ | $r_{rep}$ | $n_E$ | $n_T$ |
| AGE | 0.24 | 0.21 | 150 | 178 | 0.16 | 0.14 | 275 | 400 | 0.14 | 0.13 | 325 | 468 | 0.18 | 0.15 | 250 | 342 | – | – | – | – |
| GENDER | 0.62 | 0.64 | 25 | 15 | 0.67 | 0.68 | 25 | 13 | 0.65 | 0.68 | 25 | 13 | 0.67 | 0.68 | 25 | 13 | 0.67 | 0.68 | 25 | 13 |
| INHIBITION | 0.16 | 0.14 | 325 | 388 | – | – | – | – | 0.14 | 0.12 | 425 | 551 | – | – | 425 | 551 | 0.13 | 0.11 | 425 | 606 |
| FLUID INTELLIGENCE | 0.25 | 0.25 | 100 | 121 | 0.17 | 0.14 | 300 | 379 | 0.16 | 0.14 | 300 | 379 | 0.17 | 0.14 | 275 | 423 | 0.17 | 0.14 | 300 | 408 |
| FLUID COMPONENT | 0.17 | 0.15 | 275 | 371 | 0.13 | 0.11 | 425 | 605 | 0.15 | 0.14 | 425 | 605 | 0.14 | 0.12 | 375 | 437 | 0.14 | 0.11 | 400 | 609 |
| CRYSTAL COMPONENT | 0.26 | 0.27 | 100 | 109 | 0.17 | 0.17 | 225 | 266 | 0.19 | 0.18 | 225 | 266 | 0.18 | 0.17 | 175 | 237 | 0.18 | 0.18 | 225 | 240 |
| EARLY COMPONENT | 0.22 | 0.34 | 50 | 65 | 0.16 | 0.14 | 300 | 394 | 0.18 | 0.15 | 300 | 394 | 0.17 | 0.14 | 275 | 358 | 0.15 | 0.13 | 300 | 450 |
| TOTAL COMPONENT | 0.23 | 0.27 | 125 | 104 | 0.19 | 0.19 | 200 | 221 | 0.23 | 0.22 | 200 | 221 | 0.21 | 0.19 | 125 | 165 | 0.22 | 0.20 | 175 | 205 |
| READING RECOGNITION | 0.25 | 0.22 | 150 | 158 | 0.16 | 0.15 | 300 | 346 | 0.17 | 0.15 | 300 | 346 | 0.16 | 0.14 | 275 | 368 | 0.16 | 0.15 | 275 | 353 |
| PICTURE VOCABULARY | 0.24 | 0.24 | 125 | 131 | 0.17 | 0.15 | 275 | 346 | 0.18 | 0.17 | 275 | 346 | – | – | 225 | 285 | – | – | – | – |
| SPATIAL ORIENTATION | 0.27 | 0.24 | 100 | 132 | 0.17 | 0.16 | 250 | 330 | 0.19 | 0.16 | 250 | 330 | 0.17 | 0.15 | 225 | 315 | 0.18 | 0.16 | 250 | 309 |
| NEGATIVE AFFECT (ANGER) | 0.16 | 0.14 | 300 | 423 | 0.14 | 0.14 | 325 | 399 | 0.13 | 0.13 | 325 | 399 | 0.15 | 0.13 | 375 | 508 | 0.14 | 0.13 | 350 | 509 |
| LIFE SATISFACTION | 0.14 | 0.14 | 350 | 431 | – | – | – | – | – | – | – | – | – | – | – | – | – | – | – | – |
| ENDURANCE | 0.25 | 0.24 | 150 | 135 | 0.21 | 0.21 | 150 | 175 | 0.24 | 0.23 | 150 | 175 | 0.21 | 0.21 | 100 | 147 | 0.23 | 0.22 | 125 | 159 |
| DEXTERITY | 0.18 | 0.16 | 275 | 317 | 0.16 | 0.16 | 300 | 318 | 0.16 | 0.15 | 300 | 318 | 0.17 | 0.15 | 300 | 363 | 0.16 | 0.15 | 300 | 343 |
| STRENGTH | 0.54 | 0.49 | 25 | 30 | 0.55 | 0.56 | 25 | 22 | 0.56 | 0.56 | 25 | 22 | 0.56 | 0.57 | 25 | 22 | 0.57 | 0.57 | 25 | 21 |
| TASTE | 0.14 | 0.16 | 275 | 324 | 0.15 | 0.16 | 275 | 324 | 0.17 | 0.16 | 275 | 324 | 0.16 | 0.16 | 225 | 299 | 0.16 | 0.16 | 225 | 307 |
| WORKING MEMORY | – | – | – | – | – | – | – | – | 0.13 | 0.10 | 425 | 749 | – | – | 425 | 749 | – | – | – | – |
| SELF EFFICACY | – | – | – | – | – | – | – | – | 0.14 | 0.13 | 375 | 501 | – | – | 375 | 501 | 0.13 | 0.12 | 375 | 572 |
| AGREEABLENESS | – | – | – | – | – | – | – | – | – | – | – | – | 0.13 | 0.11 | 400 | 660 | – | – | – | – |
| WORD MEMORY | – | – | – | – | – | – | – | – | – | – | – | – | – | – | – | – | 0.13 | 0.11 | 425 | 663 |

Comparison of empirical and theoretical replicability estimates for 21 phenotypes from HCP across five DWI-connectome weighting schemes (SC, FA, RD, AD, ADC). For each phenotype, empirical effect sizes and corresponding sample sizes required for replication are shown alongside theoretically estimated sample sizes. Note: theoretical replicability estimates depend on parameters such as sample size ($n = 1000$ for theretical purposes) and significance level ($\alpha = 0.05$). The effect sizes considered for deriving theoretical sample sizes was from the replication sample. Phenotypes not listed in this table (37 out of the 58 investigated phenotypes) did not yield significant effect sizes or failed to meet replicability thresholds. Effect sizes from the replication dataset at minimum replicable sample sizes were considered to derived the theoretical effect sizes. Effect sizes from replicable sample set were considered here as the predictions in the replication dataset are less biased than that of the discovery dataset, that was used to train the models.

acquired with repetition time 2400 milli seconds (ms), echo time 2.14 ms, inversion time 1000 ms, flip angle 8°, field of view 224 × 224 mm, voxel size 0.7 mm isotropic, bandwidth 210 $H_Z/P_X$, iPAT (integrated Parallel Acquisition Techniques) 2, and acquisition time 7 min 40 s. The diffusion weighted images were acquired in 90 directions, consisting of an equal number of shells, i.e., $b = 1000$, $b = 2000$, and $b = 3000$ s/mm², along with 6 b0 images. The diffusion weighted parameters included repetition time 5520 ms, echo time 89.5 ms, flip angle 78°, refocusing flip angle 160°, field of view 210 × 180 mm, matrix 168 × 144, slice thickness 1.25 mm, 111 slices, 1.25 mm isotropic voxel, multiband factor 3, echo spacing 0.78 ms, bandwidth 1488 $H_Z/P_X$, phase partial Fourier 6/8, right-to-left and left-to-right phase encoding polarities. For AOMIC data, the detailed scanning protocol for the T1-weighted and DWI sequences can be found from the main scientific data article[7]. Whole brain 3D MPRAGE T1 images were acquired with a repetition time 8.1 ms, echo time 3.7 ms, flip angle 8°, field of view 160 × 256 × 256 mm, voxel size 1 mm isotropic, bandwidth 191.5 $H_Z/P_X$ and acquisition time 5 min 58 s. The diffusion images were acquired in 32 directions with 1 shell, i.e., DWI $b$-value 1000 s/mm², along with 1 b0 image. Repetition time was 6312 ms, echo time 74 ms, flip angle 90°, field of view 224 × 224 × 120, matrix size 112 × 112, voxel size 2 mm isotropic with a duration of 4 min 49 s.

## Behavioral phenotypes
A total of 58 behavioral phenotypes were analyzed from the HCP cohort (please see Supplementary Table 1, https://www.humanconnectome.org/, HCP_S1200_Release_Reference_Manual.pdf). The phenotypes belong to 6 different behavioral batteries namely alertness, cognition, emotion, motor, personality and sensory. Each of these domains were categorized into either trait-like or state-like phenotypes. In brief, trait-like phenotypes represented stable characteristics such as cognition, motor, sensory, emotion and personality traits. State-like phenotypes represented transient fluctuations captured by the emotion battery. The AOMIC data consisted of 19 behavioral phenotypes including fluid intelligence, memory, crystallized intelligence, behavioral activation system scales of drive, fun and reward, behavioral inhibition system, five-factorial model of neuroticism, extraversion, openness, agreeableness, conscientiousness, and trait anxiety scale[7] (see Supplementary Table 1). All the 19 behavioral measures were categorized as trait-like phenotypes.

## DWI image processing
For DWI analysis, the aim was to calculate 5 different types of connectomes, namely, streamline-based connectome, and 4 different weighted connectomes based on diffusion tensors, namely FA, radioal diffusivity, AD and ADC. Image data analyses were conducted for achieving participant level SC matrices—connectomes—which are comprised of nodes and edges. Nodes were defined as gray matter regions of interest, and edges connect a pair of nodes through white matter tractography. For the nodes, individual anatomical T1-weighted images were segmented into bilateral regions of comprising of 68 nodes based on Desikan-Killiany-Tourville atlas[20] using FreeSurfer. Diffusion weighted image analyses were conducted using MRtrix3 to generate whole brain white matter-based tractography. Tractography analyses were performed incorporating the recommendations provided in MRtrix3[21] basic and advanced tractography tutorial (https://osf.io/fkyht/). In brief, images from both phase encoding directions (right-to-left and left-to-right) were denoised (*dwidenoise*) and corrected for Gibbs ringing artefacts (*mrdegibbs*). Subsequently, corrections for motion, eddy-current and susceptibility induced distortions were applied using the *dwifslpreproc* command, which invokes FSL's eddy and topup processes. Response functions for white matter, gray matter, and cerebrospinal fluid were estimated using the dwi2response dhollander command. Fiber Orientation Distributions (FOD) within each voxel were estimated with multi-shell multi-tissue constrained spherical deconvolution (*dwi2fod msmt_csd*), utilizing the diffusion data and response functions. The FODs were then normalized using mtnormalise. The T1 image was segmented into five distinct tissue classes (cortical and subcortical gray matter, white matter, cerebrospinal fluid, and

pathological tissue) using 5ttgen and co-registered to the diffusion space with the $b = 0$ image serving as the reference. Subsequently, an interface image of the gray-white matter boundary was generated using 5tt2gmwmi. 10 million streamlines were generated with *tckgen*, based on the white matter FOD image, guided by the anatomic constraints of the 5tt image, and seeded from the GM-WM interface. These tractograms were filtered using the *tcksift2* algorithm, which normalizes the tractogram by assigning a cross-sectional area multiplier, aiming to obtain biologically accurate measures of fiber connectivity. Finally, using *tck2connectome*, DWI-based streamline (tracts) was converted to a connectome by mapping the reconstructed streamlines to a predefined brain parcellation atlas (Desikan Killiany atlas). Specifically, connectivity strength between two nodes was defined as the number of connecting streamlines scaled by the inverse of the nodal volumes. This normalization aimed to account for nodal volume-related biases in connectivity. Along with the connectivity strength, i.e., streamline-based connectivity (SC), tract-wise connectome matrices for diffusion tensor maps namely FA, RD, ADC, and AD were also calculated, which reflects the microstructural properties of white matter connection integrities. For creating DTI metric-based (FA, RD, AD, ADC) connectome, the connectome matrix was weighted by the respective metrics through a multi-step procedure. For example, for every reconstructed streamline, the average FA value was computed by sampling the underlying FA image along its trajectory. Then, as each streamline was assigned to connectome edges (connections between predefined brain regions), its contribution to that edge's weight was modulated (multiplied) by its calculated mean FA value. Finally, the strength of each edge in the connectome matrix was determined by averaging the FA-weighted contributions of all streamlines connecting the respective pair of regions. Overall, this process resulted in a total of five different types of connectivity metrices (connectomes) per participant, reflecting different aspects of the white matter structural integrity[22]. For example, FA quantifies the directionality of water diffusion, with higher values indicating greater structural organization. AD—which corresponds to the principal eigenvalue of the diffusion tensor (λ1)—measures diffusion along the primary direction of the axonal tracts, reflecting axonal integrity. RD measures diffusion perpendicular to the principal eigenvalue, sensitive to myelin integrity (λ1 + λ2)/2. ADC, otherwise known as mean diffusivity, represents the overall magnitude of water diffusion within the voxel (λ1 + λ2 + λ3)/3, indicative of general tissue characteristics.

## Replicability assessment machine learning pipeline
For the replicability analyses (see Fig. 1A *and* https://github.com/pni-lab/dwi-replicability), connectomes were considered as features, with behavioral measures which served as different targets. For each phenotypic target variable, we employed a multivariate predictive modeling approach. Specifically, we utilized the DWI-derived connectivity measures between multiple brain regions as a set of independent variables to predict a single dependent variable, representing the behavioral or phenotypic measure of interest. Therefore, while the prediction target was a single phenotype at a time, the models themselves leveraged the information contained within the high-dimensional DWI connectome, encompassing connectivity across numerous brain regions. All replicability analyses were conducted on a per-phenotype basis using this multivariate framework. Machine learning models were constructed with linear regression (Ridge/L2 regularization). We employed ridge regression for our multivariate analyses due to its effectiveness in handling the high dimensionality and multicollinearity common in brain connectivity data via L2 regularization, leading to stable and generalizable models, a frequent choice in brain-phenotype studies (e.g., refs. 2,23–26). Compared to computationally demanding non-linear methods requiring extensive tuning, ridge regression offers efficiency and interpretability, aligning with our focus on establishing a robust and replicable baseline with minimal hyperparameter optimization (solely the learning rate). Please see Supplementary Table 2 for analyses conducted using non-linear methods such as Support Vector Regression (SVR) and Kernel Ridge. Data was equally split into two samples, discovery sample (for train-test) and replication sample (validation). No explicit demographic

matching or targeted stratification was performed during this splitting process. This decision was made to assess a more realistic form of out-of-sample generalization, characterizing model replicability when applied to arbitrary, unseen subsets drawn from the same underlying population, as opposed to evaluating replication in a perfectly matched sample. The models were developed on the discovery sample using a nested cross-validation approach with 5-fold split at both outer and inner loops of the cross validation. Subsequently, the models were applied to both the discovery and replication sets for generating predictions of phenotypes, and the *p* values of both discovery and replication datasets were determined (i.e., *p* value for y_discovery and yhat_discovery, y_replication and yhat_replication). This model pipeline was repeated with 100 shuffles, for each phenotype and replication probability per phenotype was calculated at *p* < 0.05. In addition, the model results were also obtained for variable sample sizes, i.e., 25–450 in step size of 25 (equal sample size for both the discovery and replication sets, total sample size = discovery + replication samples). For a given sample size, and number of permutations, $n = 100$, replication probability was calculated as:

$$P_{\text{replicability}} = \frac{\#_{i=0}^{n}\left\{p_{\text{replication}_i} < 0.05 | p_{\text{discovery}_i} < 0.05\right\}}{\#_{i=0}^{n}\left\{p_{\text{discovery}_i} < 0.05\right\}}$$

Where octothorp # denotes the number of permutation cases satisfying condition within the curly brackets and, $p_{\text{discovery}}$ is the *p* value for the cross-validated model performance estimate during discovery, $p_{\text{replication}}$ is the *p* value of the association between model output and target phenotype in the replication sample and $P_{\text{replicability}}$ is the probability for a given behavioral model to be replicable at a given sample size. Crucially, when we state "sample sizes needed for replicable associations", we mean that the model demonstrated consistent performance when trained on n independent participants and then validated on a separate, independent set of n participants—both sets belonging to the same dataset. In other words, reported sample sizes for replicability refers to the number of discovery (training) samples needed to produce a significant effect in the replication (testing set). In our primary analysis, explicit outlier detection and rejection were not performed for either the imaging features or the target variable prior to model training. This decision was made to avoid presenting potentially overly optimistic estimates regarding sample size requirements for predictive performance. Instead of data exclusion, we relied on the ridge regression algorithm, which incorporates L2 regularization to mitigate the influence of extreme values. For comparison, results from analyses where outliers in the target variable were excluded based on a threshold of 3 standard deviations from the mean are detailed in Supplementary Table 12.

### Effect size-based theoretical sample sizes

To provide guidance on the sample size required for achieving a high probability of successful replication ($P_{\text{srep}}$), we estimated theoretical sample size needs. This estimation was based on the observed effect sizes in our replication samples and primarily assesses the univariate association between predicted outcome values and actual observed target values. This fundamental relationship is maintained even when predicted values are derived from complex methodologies like nested cross-validation and empirical *p* values are determined by permutation testing. This approach aligns with methods related to Killeen's concept of replication probability ($p_{\text{rep}}$), justifying the use of formulas based on a single effect size (e.g., Pearson's correlation *r* or $R^2$ between predicted and observed values)[27]. Utilizing the effect size that quantifies this univariate relationship as observed in the replication sample, we calculated the theoretical probability of successfully replicating this effect size ($P_{\text{srep}}$) across a range of potential future sample sizes (from 1 up to a maximum of 1000 in this analysis). Killeen's $p_{\text{rep}}$ estimates the probability that a replication study will find an effect in the same direction as the original study. An extension of this, $P_{\text{srep}}$, calculates the probability of obtaining a statistically significant result in the replication sample at a given

alpha level (α), which defines replication success. As a caution, it should be noted that while such probabilities are often evaluated against a conventional significance threshold (e.g., $\alpha = 0.05$), this specific cut-off can be considered arbitrary and may lead to an artificial dichotomy between significant and non-significant findings. According to Lecoutre et al.[27], for unknown variance, $P_{\text{srep}}$ can be calculated using the K-prime distribution:

$$p_{srep}(\alpha) = \Pr\left(t_{rep} > T_\alpha | t_{obs}\right) = \Pr[K'_{(\nu,\nu)}\left(t_{obs}/\sqrt{2}\right) > T_\alpha/\sqrt{2}]$$

where $K'$ is the K-prime distribution, $\nu$ represents degrees of freedom, tobs and trep are the *t* values from the discovery and replication samples, respectively, and $T_\alpha$ is the critical *t* value for the chosen significance level α. The *t* value for $t_{\text{obs}}$ can be derived from Pearson's correlation (r) as:

$$t_{obs} = r * \sqrt{\frac{n-2}{1-r^2}}$$

Using these formulas, the $P_{\text{srep}}$ can be determined for various potential future sample sizes based on the effect size observed. For instance, even with a modest correlation of $r = 0.1$ in a discovery sample of $n = 1000$, the probability of significant replication in a similarly sized sample is substantial. Furthermore, these equations allow for the determination of sample sizes needed to achieve a desired replication probability (e.g., 80%) for specific effect sizes. For example, to achieve 80% $P_{\text{srep}}$ with effect sizes of $r = 0.14$, $r = 0.19$, and $r = 0.32$ (corresponding to 2%, 4%, and 10% explained variance, as reported for DWI measures in ref. [13]), the required sample sizes are approximately $n = 425$, $n = 269$, and $n = 91$, respectively. These theoretically estimated sample sizes were subsequently compared to the actual sample sizes of the phenotypes included in our study (those with valid univariate effect size estimates from the prediction-outcome relationship) to assess the adequacy of current sample sizes for achieving this desired level of replication probability across varying significance thresholds.

### Additional confounder testing of predictive models

While the primary focus of this work is on the predictive yield and replicability strengths of DWI imaging-based connectome models, it is crucial to consider that several phenotypes, and consequently their predictions, can be confounded by TIV. To address this potential bias, we performed a confounder analysis using "mlconfound"[11]. This analysis involved two key aspects, (A) Partial confounder test—Assesses if predictions are partially driven by the confounder. A $p < 0.05$ indicates a significant partial influence of the confounder. (B) Full confounder test—Determines if predictions are entirely driven by the confounder. A $p < 0.05$ in this test suggests that the predictions are not solely explained by the confounder, implying that the features of interest (e.g., structural connectivities) still hold significant predictive strengths. For this analysis, three variables were utilized: (1) y, the actual behavioral target; (2) $\hat{y}$, the SC-predicted behavioral outcome; and (3) c, the TIV. To obtain yhat, the predictive model was trained and tested on the entire sample size of HCP.

### Reporting summary

Further information on research design is available in the Nature Portfolio Reporting Summary linked to this article.

### Data availability

The white matter tractography data for actual MRI data will be provided upon reasonable request.

### Code availability

The scripts related to the project are provided with a GitHub repository at https://github.com/pni-lab/dwi-replicability.

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

## Acknowledgements

The work is funded by the (DFG, German Research Foundation)—Project-ID 422744262–TRR 289 (Gefördert durch die Deutsche Forschungsgemeinschaft (DFG)—Projektnummer 422744262–TRR 289) and IFORES-Projektförderung Career Kickstart IFORES D/107-30310. Data processing was also executed on the high-performance computing (HPC) cluster of the Institute of Artificial Intelligence in Medicine (IKIM), University Medicine Essen and University of Duisburg-Essen, Germany.

## Author contributions

R.K. and T.S. conceived the study. R.K. conducted the analyses. R.K. and T.S. wrote the study. R.K., B.K., G.G., R.E., K.H., J.L., C.B., U.B., and T.S. contributed to the interpretation, as well as manuscript revision. The whole study was supervised by R.K. and T.S.

## Funding

## Competing interests

The authors declare no competing interests.
