## [Transparent Peer Review file · Communications Biology]

On the replicability of diffusion weighted MRI-based brain-behavior models

Corresponding Author: Dr Raviteja Kotikalapudi

Version 0:

Reviewer comments:

Reviewer #1

(Remarks to the Author)

In the manuscript, "On the replicability of diffusion weighted MRI-based brain-behavior models", Kotikalapudi et al quantify the multivariate replicability of diffusion weighting imaging (DWI)-based prediction of 58 phenotypes from the Human Connectome Project young adult cohort. While the methods seem straightforward, I could not find the details anywhere to make an honest assessment. The paper leads with a very short abstract that leads the reader to believe that the majority of phenotypes are replicable with small to moderate samples ($n < 500$); however, this number comes from 13 of the 58 phenotypes, which is misleading. I have several deep concerns about the paper as it is framed and reported, with those concerns detailed below.

The authors state in the abstract that 67-75% of trait-like phenotypes can be replicable with sample sizes of $n < 500$. This is a vague statement at best. What is the distinction between a trait-like and state-like phenotype? Which of the 58 phenotypes are traits and which are states? Where does mental health fall into this? Personality is certainly not a state-like phenotype. Leading with this information in the abstract, while not mentioning that overall, only ~1/3 of phenotypes are replicable with samples less than 500 (in the test set?) is misleading at best.

In the abstract the authors claim that we should prioritize stable and reliable target phenotypes. While I certainly agree that the reliability of phenotypes is of great importance, why should "research" focus on stable phenotypes? Should funding agencies and scientists not be focus on mental health symptoms that fluctuate over time (e.g., depression symptoms)?

In the Introduction, the authors state "In contrast to mass-univariate BWAS, multivariate statistical learning techniques can achieve much higher effect sizes and can therefore be replicable with low to moderate sample sizes in many cases". Aside from cognition, I am not aware of any studies showing low-to-moderate sample prediction of mental health phenotypes in a BWAS framework. The word "many" is misleading.

The authors state that they studied 58 phenotypes, but only report on 19 (as far as I can tell) in the Figure 1E,F. This is deeply misleading. Where are the results for the majority of the phenotypes?

It is not clear whether the authors are making claims about TOTAL (train + test) sample size or just training or test sample size when referring to their sample sizes needed for replicability. This matters immensely in the interpretation of the results.

The authors state in the abstract that 67-75% of trait-like phenotypes are replicable in $n < 500$ (again unclear if that is total train + test .. or just test .. or just training data), but in Figure 1d, the middle panel claims that number to only be 50%.

In panel D, it is shown that 31% of the phenotypes overall are replicable, but in panel B, the replicability percentage never gets over 31% across the DWI measures. So, how can 31% of the phenotypes overall be replicable. Are the authors just reporting the highest value?

I cannot make any statements on the rigor on the rigor of the multivariate methods here, as there is no Methods section in the main document or supplement.

Can the authors provide a table that lists all the 58 phenotypes?

Reviewer #2

(Remarks to the Author)

This manuscript presents a very timely examination of the replicability of multivariate analyses of diffusion data and various behavioral and demographic phenotypes across a range of sample sizes. The approach is rigorous with respect to the HCP dataset, however one is left wondering how the results might generalize to other datasets with different diffusion acquisitions. Further, while the number and extent of the replication analyses conducted is impressive, they are limited to a rather narrow focus on ridge regression as a multivariate modeling approach and the Desikan-Killiany atlas. If the authors insist on limiting analyses to these approaches, additional justification and rationale should be provided for these crucial decisions. I expand on these comments as well as additional areas where clarity and impact of this strong work can be improved below.

1. While I understand this is a publicly available dataset, there should still be basic information about the dataset included in the paper. For example, the total sample size used in the paper is not reported. Additionally, the paper sometimes refers to $n < 500$ and sometimes $n \leq 450$. It does not appear there were any permutations that included 500 discovery and replication observations, so this figure should be edited.
2. Related, there is no information provided on whether/how quality assessments were performed for the diffusion data or for the phenotype data. Did the final included sample pass any sort of QA? This is critical to understand.
3. When data were subsampled into discovery and replication sets, was there any effort to demographically match each set or to ensure similar distributions of key variables across discovery and replication? The decision making here should be communicated in the paper. This relates to the authors open-ended discussion of generalizability across sub-populations. Can this be addressed in the current dataset?
4. Related to generalizability, the results presented here could be much more impactful if the authors could include a second dataset to identify how the reported sample size-replicability relationships hold across datasets with different imaging acquisitions and/or more diverse samples (e.g. datasets like the Philadelphia Neurodevelopmental Cohort which is single-shell and has demographics representing the Philadelphia community; of course many other datasets exist as well).
5. The authors use ridge-regression for multivariate analyses. Do they expect this to be consistent with other multivariate approaches (e.g. SVM, random forest, etc)? The authors should at least provide some rationale for this choice to inform readers.
6. The Desikan-Killiany atlas was used to create connectomes. This is a fairly coarse parcellation. How did the authors choose this atlas, and might the results change if a higher-resolution atlas was used that generated more image features for the multivariate model?
7. It is unclear exactly how connectivity strength, referred to as SC, was calculated.
8. The authors evaluate the likelihood of obtaining results of $p < .05$ across replication and discovery. This is certainly one definition of replicability, but another (which perhaps is even more relevant) is how much the effect itself replicates across datasets. The authors should also compare the magnitude of effect size across sample sizes to elucidate this key property.

Minor points:

1. Third paragraph of Main text: R2 values are listed as percentages (e.g. 51.7%) when they should be provided as decimals (e.g. .517).
2. Figure caption: typo "guven"  "given".

Reviewer #3

(Remarks to the Author)

Brief summary of the manuscript

In this manuscript, Kotikalapudi et al. evaluate the replicability of using diffusion-weighted imaging (DWI) data to predict trait-like phenotypes. The DWI measures analyzed include structural connectivity (SC), fractional anisotropy (FA), radial diffusivity (RD), axial diffusivity (AD), and apparent diffusion coefficient (ADC). The study examines 58 behavioral and psychometric phenotypes derived from the Human Connectome Project. Through 100 permutations with varying participant sample sizes, the authors estimated the effect size using the coefficient of determination R^2 and employed Killeen's formula as a replicability metric. They concluded that using DWI data for prediction is challenging for traits related to emotions and personality but feasible with a sample size of a few hundred participants for phenotypes related to cognition, motor skills, and demographics.

Overall impression of the work

This study is contextualized within the growing body of research on Brain-Wide Association Studies (BWAS), which has been argued to require sample sizes in the thousands, as suggested by previous work such as Marek et al. (2022). The

authors use established methods—such as train-test splitting, regularization, permutation, and Killeen’s probability to assess replicability—and provide their Python code openly on GitHub.

The technical rigor of the study could be improved. While the authors’ optimistic conclusion regarding sample size requirements for some phenotypic traits is noteworthy, it relies heavily on the robustness of their methodology, which requires closer scrutiny. This raises concerns about the strength and validity of their conclusions. Furthermore, more detailed descriptions of the models used would improve transparency and help prevent potential misinterpretations.

Specific comments

A. Effect Size and Statistical Significance

1) The replicability in this study is measured using Killeen’s replication probability, which estimates the likelihood that a significant effect size observed in the training dataset will be replicated in the testing dataset. However, Killeen’s replication probability is based on a significance threshold of 0.05, an arbitrary cut-off that can lead to an artificial dichotomy between significant and non-significant results.

2) In the supplemental material, the authors present the formula for the t-statistic of the Pearson correlation coefficient r , which is used to assess statistical significance:

$$t = r * \text{sqrt} \left((n - 2) / (1 - r^2) \right)$$

Here, the adoption of the term $n - 2$, also representing the degrees of freedom for the t-statistic, assumes simple linear regression, where the slope and intercept are estimated. However, in the context of this study, which involves many predictors and regularization, directly applying a t-test, as defined above, may not be appropriate. The complexities introduced by regularization and multiple predictors need to be carefully considered when assessing statistical significance.

3) The definition of replicability used here focuses solely on statistical significance (i.e., whether R^2 is significantly different from zero), without sufficient attention to effect size. This could lead to misinterpretation, particularly when small effect sizes are found to be significant. While the authors note in the supplemental material that even a small R^2 (e.g., a few percent) can yield reasonable replicability with a couple hundred participants, the practical implications of such small effects should be carefully evaluated, especially when larger effects are desired for clinical or real-world relevance.

B. Multivariate vs. Univariate Modeling

The manuscript frequently references multivariate modeling. However, it is unclear whether each phenotypic trait is predicted individually using DWI data. If the models predict one trait at a time, the approach would be more accurately described as univariate rather than multivariate. Clarifying this distinction would help avoid confusion. It would be helpful to explicitly present the regression models in the supplemental material. In addition, to enhance understanding and transparency, it should be explained how the regression is handled in cases where binary and discrete traits (e.g., sex or personality traits) are involved.

C. Prediction vs. Mechanism

While the primary objective of BWAS is to make predictions based on neuroimaging measures, there is an inherent focus on association, as suggested by the term itself. The authors rightly emphasize that “it is important to consider whether the phenotype of interest is relevant at all to the investigated brain measure.” Although association analyses can identify relationships between variables, they do not imply causation.

To provide deeper insights, it would be valuable for the authors to discuss which specific brain regions or networks are particularly associated with each phenotypic trait (e.g., sex, age, cognition, motor skills). Understanding these associations could help bridge the gap between prediction and mechanistic understanding, potentially guiding future studies that aim to uncover causal links.

Version 1:

Reviewer comments:

Reviewer #1

(Remarks to the Author)

The manuscript has improved substantially since the initial submission. I appreciate the work the authors have made to reply to my comments. Given all of the updates that were made, I now have a few more comments that I would like to authors to attend to.

1. Did the authors control for intracranial volume in any way? I understand the prediction (as is the focus here) is different from trying to draw an inference about the SC-phenotype association. However, I could see intracranial volume, which varies as a function of known confounds/colliders (e.g., SES), acting as a hidden confound in the prediction of these

phenotypes from structural data.

2. I like the addition of Figure 1C. It is very informative and a nice way to display a lot of data. What strikes me is the pattern of putative replicability in more moderate samples. Age, gender, cognition are the apparent winners. However, as mentioned in the previous point, how much of this could be driven by a hidden confound like head size or SES? Said differently, it is not the brain per se predicting gender or cognition, but rather the phenotype of interest varies with the confound and thus the confound is being aliased into the brain prediction.

3. Overall, 36% of phenotypes were replicable, but replicability in Figure 1B is defined as replicable at an $n \leq 425$ in at least one SC metric. That seems to be a low bar. A better representation of the data would be the proportion of phenotypes that are replicable in at least one, at least two, three, four, and all 5 SC metrics. The authors somewhat get at this in Supplementary table 3. However, it's not clear if, for example, the 16 phenotypes that are replicable with ADC are completely overlapping with RD. My guess, based on Supplementary Figure 2, is that there is a lot of overlap, but it would be good to detail. Likewise, for the heatmap in Figure 1C, is this showing the probability of replication for one of the SC metrics? Or just the best performing for each phenotype?

4. The authors should make it clear in the text that when an $n < 425$ is referenced that is assuming an n of 425 in both the train and test set of data. As such, these estimates are based on a total sample size of 850, for example. However, it should not be assumed that a model will replicate/generalize equally well to a brand-new sample of smaller N (e.g., 50). The language used around this topic throughout the abstract and main text is ambiguous. The authors should either make it clear than the sample refers to the training sample size or (more preferably) make it crystal clear that their estimates are based off a total sample size twice as big as what they are listing. This is especially important as quality reference samples used to benchmark replicability is dependent on its size.

Reviewer #2

(Remarks to the Author)

The authors have greatly improved the manuscript with this revision by expanding their analyses, providing additional effect size information, and adding an additional dataset. The new Table 1 is itself a major improvement to the description and clarity of the findings. That said, I do have a couple of lingering concerns, which I consider to be minor.

1. The primary metric highlighted, and the only one used in the second AOMIC dataset, is streamline count (SC). My concern is that SC is arguably the least quantitative of the available measures, in the sense that it is the least likely to be numerically stable across different tractography processing decisions. The number of streamlines may depend heavily on the seeding scheme (including the random seed used, the number of seeds placed, etc.), as well as on whether tractography is constrained by the number of placed seeds or the total number of streamlines derived. It is possible that the authors would recommend that future studies use the exact tractography procedures employed here, but that feels overly narrow. I do understand that SC performed best in the presented analyses, but given its potential limitations, this warrants more discussion. It would also be helpful to see how other metrics, such as FA, generalize to the new dataset. Regardless of the outcome, this would be informative (if other measures also generalize well, that supports broader applicability; if not, it would help clarify the specificity of the findings).

2. In Figure 1, panels D, E, and F appear to indicate that trait and state measures will be represented in black vs. purple based on the text colors, but that is not reflected in the actual figure panels. Additionally, in panel C, the blue color scale represents P-replicability, whereas in panel E it represents sample size. It is unclear what blue represents in panels D and F. Overall, the use of color in this figure could be improved to make this sole main text figure more intuitive and easier to interpret.

Reviewer #3

(Remarks to the Author)

The authors have responded thoroughly and thoughtfully to the reviewers' comments. I am satisfied with the revisions and commend the authors for their careful attention to detail.

I have just one minor grammatical suggestion. In the following sentence, the first "the" appears to be a typographical error and should likely be replaced with "of":

"Our primary replicability analyses on the HCP dataset utilized the 84-region Desikan-Killiany brain atlas, which is the default option the the software FreeSurfer."

The corrected sentence should read:

"...which is the default option of the software FreeSurfer."

Overall, I find the revised manuscript suitable for publication, subject to this minor correction and the final evaluation by the editors and other reviewers.

Version 2:

Reviewer comments:

Reviewer #1

(Remarks to the Author)

The authors have sufficiently addressed all of my concerns.

On the replicability of diffusion weighted MRI-based brain-behavior models – Response to reviewers

Dear Editor-in-chief and respected reviewers,

We would like to express our sincere gratitude for taking the time to review our manuscript. We have received valuable feedback from all three reviewers, which has been instrumental in improving our work. Our response and manuscript updates have addressed each of the points raised by the reviewers, and we are confident that this has enhanced the quality of our research. Key among these revisions is the replication of our results on an independent large-scale DWI-MRI dataset, alongside explorations using a higher resolution atlas to test spatial robustness, various machine learning algorithms to examine methodological generality and a better representation of empirical versus theoretical replicabilities. The manuscript's overall format has also changed. While it was originally submitted as a "short communication" we have now significantly expanded it to include the requested details on methods and results. This has allowed us to support our claims in a more detailed and cohesive manner, while remaining relatively concise and in line with the journal's formatting guidelines. During the revision, we received valuable assistance from our colleague, Dr. Christian Büchel, who is now listed as a co-author on the manuscript.

We hope that the revisions and our accompanying responses will be sufficient to make our manuscript suitable for publication in *Communications Biology*. We would be also glad to respond to any further questions and comments that you and the referees may have.

We are looking forward to hearing from you.

Sincerely,

Dr. Kotikalapudi and Prof. Tamas Spisak, on behalf of all the authors

Reviewer 1

R1C1. In the manuscript, “On the replicability of diffusion weighted MRI-based brain-behavior models”, Kotikalapudi et al quantify the multivariate replicability of diffusion weighting imaging (DWI)-based prediction of 58 phenotypes from the Human Connectome Project young adult cohort. While the methods seem straightforward, I could not find the details anywhere to make an honest assessment. The paper leads with a very short abstract that leads the reader to believe that the majority of phenotypes are replicable with small to moderate samples ($n < 500$); however, this number comes from 13 of the 58 phenotypes, which is misleading. I have several deep concerns about the paper as it is framed and reported, with those concerns detailed below.

We thank the reviewer for the feedback. In the revised manuscript, we have abandoned the original “short communication” format and present the manuscript as a full research paper, allowing us to provide the necessary details and address the reviewer’s concerns. Among others, we have significantly extended our methods section. The abstract is also expanded to yield a more accurate summary of the findings and to avoid overinterpretation. Our actions taken to address other concerns about framing and reporting are detailed below.

2. The authors state in the abstract that 67-75% of trait-like phenotypes can be replicable with sample sizes of $n < 500$. This is a vague statement at best. What is the distinction between a trait-like and state-like phenotype? Which of the 58 phenotypes are traits and which are states? Where does mental health fall into this? Personality is certainly not a state-like phenotype. Leading with this information in the abstract, while not mentioning that overall, only ~1/3 of phenotypes are replicable with samples less than 500 (in the test set?) is misleading at best.

Indeed, we concur that cautious interpretation of the results is essential due to the inherent continuum between state-like and trait-like behaviors, meaning categorization isn't always clear-cut. We revised our categorization to follow a precise definition and made sure that its presentation is toned down with more emphasis on the overall replicability estimates (see e.g. the new **long format abstract**). Specifically, we differentiate between enduring and relatively stable personality dispositions, which represent long-lasting traits, and transient/fluctuating, short-lived feelings, which constitute temporary states (Geiser et al., 2017) (we now added this piece of information in the *main figure caption*). To provide clarity, we now provide **supplementary table 1** (Behavioural phenotypes from the HCP dataset) and a newly included dataset called the Amsterdam Open MRI Collection. Regarding the state-like nature of personality, we also revised the corresponding paragraph in the discussion to address the comment, while keeping the important discussion about the obvious inherent limitations of DWI to capture quickly changing variables:

“In contrast, phenotypic measures related to emotions (accounts to ~45% of all phenotypes in ~~our study~~ HCP1200 dataset and no state-like in AOMIC dataset) and, to some degree even personality traits of openness and neuroticism, are known to be influenced by a variety of dynamic factors, from belief updates to contextual information or biological rhythms and cycles (Sharot et al., 2023; Thornton, Weaverdyck, Mildner, et al., 2019; Thornton, Weaverdyck, & Tamir, 2019), and may exhibit relatively rapid changes over time (Geiser et al., 2017). The inherent static information representation via tractography might not be sufficient for explaining greater effect sizes in such phenotypes. DWI is inherently ill posed to predict such phenotypes, explaining the observed poor replicabilities (5-10%).”

We acknowledge the reviewer's point about mental health. While certain phenotypes in our study are relevant to mental disorders, a comprehensive evaluation specifically focused on the replicability of mental health-related clinical outcomes, e.g. diagnoses, was beyond the scope of our current investigation. This is now made explicit in our discussion on trait-state-predictive modeling following this comment. Please kindly read the immediate next comment (R1C3).

3. In the abstract the authors claim that we should prioritize stable and reliable target phenotypes. While I certainly agree that the reliability of phenotypes is of great importance, why should “research” focus on stable phenotypes? Should funding agencies and scientists not be focus on mental health symptoms that fluctuate over time (e.g., depression symptoms)?

We wholeheartedly agree that research on mental health (and medical research in general) cannot ignore state-like symptom dynamics that fluctuate over time. We only intended to argue that tractography based DWI-connectomes may not be well positioned to predict such rapid fluctuations, simply as the inherently slow rate of change/plasticity in white matter microstructure cannot reflect such fast dynamics. We acknowledge that our argumentation was not clear enough in the previous version. We have revised the abstract and expanded our discussion to provide a clearer and more balanced presentation of the topic. The **discussion** now reads,

“Phenotypes that are not only highly relevant to the brain feature of interest (e.g. trait measures for DWI) but also display a high test-retest reliability can be expected to display the highest effect sizes and will be replicable with small sample sizes. This has been exemplified by several previously reported multivariate neural signatures that showed strong, replicable effect sizes with various behavioural variables, even though they were trained on small samples, sometimes fewer than 50 participants (Spisak et al., 2020; Wager et al., 2013). We argue that such models are the most promising candidates for individual-level predictions guiding clinical care and personalized medical approaches and our findings hint that the effect sizes and replicability necessary for these applications is not impossible to achieve, at least for the most reliable, and stable or slowly changing phenotypes. This does not imply that only trait-like phenotypes are suitable for predictive modelling in general. However, for more dynamic, state-like phenotypes (like emotional state), approaches leveraging neural processes with faster dynamics, like fMRI, EEG, and MEG, are more appropriate than DWI, which is inherently limited by the slow rate of change/plasticity in white matter microstructure. For example, while the general tendency towards anxiety is a trait-like characteristic, the intensity of anxiety experienced in response to a specific social situation is a state-like attribute that fMRI task-based brain-derived phenotypes could explain more reliably than structural models. Given that mental health often involves a combination of trait-like predispositions and state-like fluctuations, it is important to note that achieving a more comprehensive understanding, diagnosis, and treatment in this domain often requires predictive performance based on both types of variables.”

4. In the Introduction, the authors state “In contrast to mass-univariate BWAS, multivariate statistical learning techniques can achieve much higher effect sizes and can therefore be replicable with low to moderate sample sizes in many cases”. Aside from cognition, I am not aware of any studies showing low-to-moderate sample prediction of mental health phenotypes in a BWAS framework. The word “many” is misleading.

We have provided reference examples (He et al., 2022; Spisak et al., 2023; Spisak et al., 2020; Wager et al., 2013; Woo et al., 2017) of multivariate models that were trained on small-to-moderate samples and yield robust (externally validated) predictive performance for phenotypes beyond cognition. Nevertheless, we agree that there are many mental health phenotypes that seem to be hard to predict even with multivariate methods, thus we have changed “many” to “several” (l.56).

5. The authors state that they studied 58 phenotypes, but only report on 19 (as far as I can tell) in the Figure 1E,F. This is deeply misleading. Where are the results for the majority of the phenotypes?

While on other parts of the same figure (Figure 1 B, C and D) we have clearly reported the number/percentage of phenotypes not replicable, we agree that the focus on replicable phenotypes was unproportional and potentially misleading at certain locations, including Fig. 1 E and F. We have improved the balance at many places, including the **abstract (completely rewritten for a long format)**, a new **Figure 1C, E, G** and added **supplementary figures 2-3 and supplementary tables 5-9** to further elaborate the findings. We thank the reviewer for the valuable feedback.

6. It is not clear whether the authors are making claims about TOTAL (train + test) sample size or just training or test sample size when referring to their sample sizes needed for replicability. This matters immensely in the interpretation of the results.

We coherently report the sample size that is needed to construct a replicable model, that is, the sample size of the discovery sample, excluding the replication sample itself. This is arguably a more informative number, in-line with the common practice that replication is conducted in a separate study. We now explicate this crucial information via **supplementary figure 1) Replicability analyses pipeline**, and also mention it in the result, main figure caption, and **methods** (see e.g. **line 76, figure 1 caption A, 341**). We have also carefully reviewed related wording to avoid misinterpretation (e.g. “sample size needed to identify replicable associations”, as opposed to saying “replicable with n=500”).

7. The authors state in the abstract that 67-75% of trait-like phenotypes are replicable in $n < 500$ (again unclear if that is total train + test .. or just test .. or just training data), but in Figure 1d, the middle panel claims that number to only be 50%.

We thank the reviewer for this correction. We have now corrected the numbers. The % replicable trait-like phenotypes were indeed 50%, i.e., 16 out of 32 phenotypes. We have now removed 67-75% also from the abstract, which is the range of replicable trait-like phenotypes in the cognition and motor domains.

8. In panel D, it is shown that 31% of the phenotypes overall are replicable, but in panel B, the replicability percentage never gets over 31% across the DWI measures. So, how can 31% of the phenotypes overall be replicable. Are the authors just reporting the highest value?

We sincerely thank the reviewer and apologize for the misunderstanding. By overall, we meant combined replicability across all DWI metrics, so that replicability with at least one of all metrics counts as a successful replication (conveying the idea to evaluate replicability for DWI in general, regardless of analysis technique). Combined replicability is often higher, because some phenotypes that are not replicable with one metric (e.g. FA), are at times replicable with another metric (SC). We apologize for the lack of clarity regarding this important information. In the revised manuscript we make this explicit, both in the **results section on Performance of DWI-connectomes with HCP dataset** and the **caption and heading of Figure 1B**. For your convenience, the caption reads:

“Overall, here, refers to considering a brain-behaviour model replicable, provided it was replicable with at least one of the 5 connectome types (SC|FA|RD|AD|ADC).”

Results section reads *“When considering successful replication as being replicable in case of at least one DWI metric, 36% (21/58) of phenotypes were replicable.”*

9. I cannot make any statements on the rigor on the rigor of the multivariate methods here, as there is no Methods section in the main document or supplement.

*We respectfully note that we did provide a methods section in the original manuscript, positioned right after the References in the main document (the references of the main text and methods had to be separated due to journal guidelines). We acknowledge that the positioning may have been confusing and apologize for the inconvenience caused to the reviewer. In the revised manuscript (also in line with requests from other reviewers), we have expanded the **methods section with more detail on the imaging protocols, behavioural phenotypes, image processing, replicability assessment through machine learning pipeline**, which along with the **supplementary information 1** on analytical sample size calculations, give a more comprehensive understandings of the technical considerations of this study.*

10. Can the authors provide a table that lists all the 58 phenotypes?

We have now made a **supplementary table 1**, for all the phenotypes pertaining to old and new analyses (please also see R1C2).

Reviewer #2

1. This manuscript presents a very timely examination of the replicability of multivariate analyses of diffusion data and various behavioral and demographic phenotypes across a range of sample sizes. The approach is rigorous with respect to the HCP dataset, however one is left wondering how the results might generalize to other datasets with different diffusion acquisitions. Further, while the number and extent of the replication analyses conducted is impressive, they are limited to a rather narrow focus on ridge regression as a multivariate modeling approach and the Desikan-Killiany atlas. If the authors insist on limiting analyses to these approaches, additional justification and rationale should be provided for these crucial decisions. I expand on these comments as well as additional areas where clarity and impact of this strong work can be improved below.

Thank you for acknowledging the timeliness and rigor of our study. We agree that our initial choice of dataset, modelling approach and brain atlas limited the generalizability and scope of our claims. We have now significantly expanded the scope of our analysis. Most importantly, we have included an additional dataset, i.e., AOMIC (<https://openneuro.org/datasets/ds003097>, n=928) with 19 behavioral phenotypes, and found that the new results align well with the original results based on HCP data. Furthermore, to move beyond the scope of ridge regression and the Desikan-Killiany atlas, we implemented additional analyses using kernel ridge regression, support vector regression, and a higher-resolution brain atlas (Destrieux atlas, 162 regions). We found that while changing the machine learning model did not seem to yield any significant improvement upon our standard pipeline, the higher resolution atlas did improve predictive performance and thus, replicability (instead of 42%, 47% of phenotypes replicated in the AOMIC dataset with $n \leq 450$, see **results second last section on exploratory analysis for improved performance using a higher-resolution atlas**). These deliberate extensions provide a more comprehensive and robust evaluation of multivariate diffusion data analyses. We are confident that these additions have strengthened the study and enhanced the generalizability of our findings.

2. While I understand this is a publicly available dataset, there should still be basic information about the dataset included in the paper. For example, the total sample size used in the paper is not reported. Additionally, the paper sometimes refers to $n < 500$ and sometimes $n \leq 450$. It does not appear there were any permutations that included 500 discovery and replication observations, so this figure should be edited.

We thank the reviewer for the valuable feedback. We have now extended the **method** section on **magnetic resonance imaging protocol (line 241 onwards)** on the newer AOMIC dataset and added a section called **behavioural phenotypes** with more dataset related details (also now included is **supplementary table 1 on behavioural measures**). We have adjusted the reporting of sample sizes throughout the manuscript. We always used a maximum of 425 samples out of 900 (17 different sample sizes from 25 to 425 in steps of 25), split equally into discovery and replication sets for both HCP and AOMIC datasets. As requested, the figures are accordingly corrected. For example, the behavioural phenotypes section reads:

*“A total of 58 behavioural phenotypes were analyzed from the HCP cohort (please see **supplementary table 1**, https://www.humanconnectome.org/HCP_S1200_Release_Reference_Manual.pdf). The phenotypes belong to 6 different behavioural batteries namely alertness, cognition, emotion, motor, personality and sensory. Each of these domains were categorized into either trait-like or state-like phenotypes. In brief, trait-like phenotypes represented stable characteristics such as cognition, motor, sensory, emotion and personality traits. State-like phenotypes represented transient fluctuations captured by the emotion battery. The AOMIC data consisted of 19 behavioural phenotypes including fluid intelligence, memory, crystallised intelligence, behavioural activation system scales of drive, fun and reward, behavioural inhibition system, five-factorial model of neuroticism, extraversion, openness, agreeableness, conscientiousness, and trait anxiety scale⁷ (see **supplementary table 1**). All the 19 behavioural measures were categorized as trait-like phenotypes.”*

3. Related, there is no information provided on whether/how quality assessments were performed for the diffusion data or for the phenotype data. Did the final included sample pass any sort of QA? This is critical to understand.

As we wanted to avoid making overly optimistic claims about sample sized needed for identifying replicable associations, we aimed to refrain from any excessive QA in our original analysis. Given the high quality and single-site nature of the HCP1200 dataset and AOMIC (mriqc was previously done for T1 and DWI images(Snoek et al., 2021)), we ended up not excluding any participants due to quality issues (or due to being an outlier) and, instead relied on the ridge regression algorithm's inherent ability (L2 penalty) to mitigate the effect of noisy features through regularization. We now clearly state this decision in the manuscript. Furthermore, following the reviewer's suggestion, now we did implement a supplementary analysis with 3SD outlier rejection at the target (phenotype) level and repeated the entire replicability analysis across all the models (SC, FA, RD, AD and ADC, please see **supplementary table 12. Effects of outliers on replicability analysis**). The results were comparable (see below). Our **method's section on replicability assessment machine learning pipeline** now reads:

*“In our primary analysis, explicit outlier detection and rejection were not performed for either the imaging features or the target variable prior to model training. This decision was made to avoid presenting potentially overly optimistic estimates regarding sample size requirements for predictive performance. Instead of data exclusion, we relied on the ridge regression algorithm, which incorporates L2 regularization to mitigate the influence of extreme values. For comparison, results from analyses where outliers in the target variable were excluded based on a threshold of 3 standard deviations from the mean are detailed in **Supplementary Table 12.**”*

Supplementary table 12. Effects of outliers on replicability analysis.

	SC (mean replicable sample size, mean r)	FA (mean replicable sample size, mean r)	RD (mean replicable sample size, mean r)	AD (mean replicable sample size, mean r)	ADC (mean replicable sample size, mean r)	Overall replicability
Original results	29 (171, .25)	26 (243, .22)	31 (240, .20)	26 (263, .21)	28 (243, .21)	21/58
After outlier correction	31 (189, .25)	28 (250, .21)	29 (253, .21)	26 (263, .22)	28 (235, .21)	19/58

4. When data were subsampled into discovery and replication sets, was there any effort to demographically match each set or to ensure similar distributions of key variables across discovery and replication? The decision making here should be communicated in the paper. This relates to the authors open-ended discussion of generalizability across sub-populations. Can this be addressed in the current dataset?

Thank you for highlighting this important detail. Like with the previous question, with this we also followed a conservative strategy, to avoid deriving replicability estimates that do not reflect the common practice in the literature. That is, we did not perform any matching between the discovery and replication sets. In the revised manuscript we explicate this decision and explain that in our opinion this approach represents a reasonable and practical balance between an extensively matched replication sample (no out-of-sample generalizability needed) and testing generalization in an entirely different population (e.g. patients, elderly, children). Note, furthermore, that for samples larger than a few hundred, such sampling balance likely becomes insignificant. For example, like the results observed in HCP datasets, we also observe similar trait-like replication in the AOMIC dataset (please see comment 5). The **method’s section on replicability assessment machine learning pipeline** now reads:

“Data was equally split into two samples, discovery sample (for train-test) and replication sample (validation). No explicit demographic matching or targeted stratification was performed during this splitting process. This decision was made to assess a more realistic form of out-of-sample generalization, characterizing model replicability when applied to arbitrary, unseen subsets drawn from the same underlying population, as opposed to evaluating replication in a perfectly matched sample.”

5. Related to generalizability, the results presented here could be much more impactful if the authors could include a second dataset to identify how the reported sample size-replicability relationships hold across datasets with different imaging acquisitions and/or more diverse samples (e.g. datasets like the Philadelphia Neurodevelopmental Cohort which is single-shell and has demographics representing the Philadelphia community; of course many other datasets exists as well).

The reviewer raises a crucial point regarding the generalizability of our findings to datasets with differing imaging acquisitions and population demographics. Thus, we have repeated our whole streamline-based analysis on another dataset, the AOMIC dataset (Amsterdam Open MRI Collection, n=928). This dataset offers a valuable contrast to the HCP, employing a single-shell diffusion acquisition protocol compared to HCP's higher resolution multi-shell sequence. Furthermore, AOMIC includes several overlapping phenotypic measures with HCP, such as cognitive measures, personality traits, age, and sex. The expanded results section and updated main figure now present these AOMIC findings (**figure 1G**), demonstrating that, encouragingly, trait-like variables show a comparable range of replicability in the AOMIC dataset to what we observed in the HCP. New section has been added to the **results** called **streamline connectivity validity with AOMIC dataset**.

6. The authors use ridge-regression for multivariate analyses. Do they expect this to be consistent with other multivariate approaches (e.g. SVM, random forest, etc)? The authors should at least provide some rationale for this choice to inform readers.

We chose ridge regression for our multivariate analyses due to its computational efficiency, efficacy in handling the inherent high dimensionality and multicollinearity of brain connectivity data without extensive hyperparameter optimization, and its widespread use in the literature. While non-linear methods like SVM and Random Forests might capture more complex relationships, they often necessitate a more extensive hyperparameter tuning process, are computationally demanding and often doesn't seem to yield more accurate predictions (Cui & Gong, 2018; Dadi et al., 2019; Pervaiz et al., 2020). We justified our default choice in the revised Methods section (see below) and added a supplementary analysis (supplementary table, see below), where we contrast the overall performance of ridge regression against non-linear kernel methods of support vector regression and kernel ridge. Overall, we found that the effect sizes and replication sample sizes were comparable between ridge regression and SVR. Kernel ridge method showed the highest variability across methods, for example replicating only 3 phenotypes in DWI-RD networks. We have incorporated the valuable comment as part of the method's section and extended analyses from supplementary informations. In **method** section on **replicability assessment machine learning pipeline**, it now reads:

“We employed ridge regression for our multivariate analyses due to its effectiveness in handling the high dimensionality and multicollinearity common in brain connectivity data via L2 regularization, leading to stable and generalizable models, a frequent choice in brain-phenotype studies (e.g., (Hoerl & Kennard, 1970; Massett et al., 2023; Spisak et al., 2023)). Compared to computationally demanding non-linear methods requiring extensive tuning, ridge regression offers efficiency and interpretability, aligning with our focus on establishing a robust and replicable baseline with minimal hyperparameter optimization (solely the learning rate). Please see supplementary results (table 2) for analyses conducted using non-linear methods such as Support Vector Regression (SVR) and Kernel Ridge.”

Supplementary table 2. Performance of non-linear kernel methods of support vector regression and kernel ridge against ridge regression.

	SC replicability % (median replicable sample size, median r)	FA replicability % (median replicable sample size, median r)	RD replicability % (median replicable sample size, median r)	AD replicability % (median replicable sample size, median r)	ADC replicability % (median replicable sample size, median r)	Overall replicability %
Ridge	29 (150, .24)	26 (275, .16)	31 (275, .15)	26 (250, .15)	28 (288, .15)	21/58
SVR	28 (150, .22)	24 (275, .15)	28 (275, .15)	24 (275, .15)	24 (275, .15)	18/58
Kernel Ridge	29 (150, .21)	33 (300, .15)	3 (138, .63)	14 (163, .19)	9 (250, .19)	20/58

Like the ridge regression, kernel ridge was tuned for the learning rate alpha with values = [1e-4, 1e-3, 1e-2, 0.1, 1e+2, 1e+3, 1e+4] and the kernel was kept at 'rbf' (radial basis function). For SVR, hyperparameter tuning was performed for the regularization strengths with values same as alpha from the other regression models.

7. The Desikan-Killiany atlas was used to create connectomes. This is a fairly coarse parcellation. How did the authors choose this atlas, and might the results change if a higher-resolution atlas was used that generated more image features for the multivariate model?

We thank the reviewer for these suggestions. On the AOMIC dataset (which we included now), we have also performed the replicability analysis with two different atlases, i.e., one with 82 nodes and another with a higher resolution atlas with 162 nodes (Destrieux atlas). The results section is also **added new results on exploratory analysis for improved performance using a higher-resolution atlas**. We also now provide **supplementary table 11** – for the same purpose (please see below). Our findings indicate that employing a higher resolution atlas could potentially enhance predictive model performance by providing richer feature information. A high-resolution atlas implementation might result in an ~5% lesser samples needed for replicability and 5% more replicable phenotypes. This informs that our overall replicability estimates are rather on the conservative side (as intended) and not inflated by methodology fine-tuned for the datasets at hand.

Supplementary table 11. AOMIC dataset – Comparison of replication using higher resolution atlas with 162 brain regions (Destrieux atlas) versus standard atlas with 82 regions (mrtrix3 labelconvert).

	Desikan-Killiany (82 regions)	Destreux (164 regions)	Desikan-Killiany	Destreux
Phenotype	Sample size for replication (n)	Sample size for replication (n)	Replicable effect size (Pearson's r)	Replicable effect size (r)
Age	250	250	.167	.167
Sex	25	25	.474	.530
BMI	200	200	.188	.209
IST_fluid	225	200	.164	.201
IST_memory	-	375	-	.128
IST_crystallised	150	100	.213	.219
IST_intelligence_total	225	125	.187	.214
BIS	425	375	.116	.130
NEO_A	325	300	.137	.144

AOMIC dataset – Comparison of replication results using a higher-resolution atlas (162 brain regions) versus a standard atlas (82 regions, post-MRtrix3 labelconvert). For each phenotype, the sample size required for replication and the replicable effect size (Pearson's r) are reported for both the standard and high-resolution atlases. Bold values indicate better performance. Notably, increasing the atlas resolution results in higher sample sizes and effect sizes, suggesting that our results from the HCP1200 cohort, based on the 82-region atlas, may be conservative. Future studies using higher-resolution atlases could potentially observe larger effect sizes, with our model findings serving as a conservative baseline for further exploration.

8. It is unclear exactly how connectivity strength, referred to as SC, was calculated.

We apologize for not being detailed enough about this in the original manuscript (which was meant to adhere to the “short communication” publication format). We have now added the necessary details in our expanded **methods (image processing)** section. We have also expanded on how other connectomes were calculated (streamline count, FA, RD, AD, ADC). Please see the **image processing section of the methods** part of the manuscript for a detailed extension on methods. For example, the image processing section now also reads:

“Finally, using tck2connectome, DWI-based streamline (tracts) was converted to a connectome by mapping the reconstructed streamlines to a predefined brain parcellation atlas (Desikan Killiany atlas). Specifically, connectivity strength between two nodes was defined as the number of connecting streamlines scaled by the inverse of the nodal volumes. This normalization aimed to account for nodal volume-related biases in connectivity. Along with the connectivity strength i.e., streamline-based connectivity (SC), tract-wise connectome matrices for diffusion tensor maps namely fractional anisotropy (FA), radial diffusivity (RD), apparent diffusion coefficient (ADC) and axial diffusivity (AD) were also calculated, which reflects the microstructural properties of white matter connection integrities. For creating DTI metric-based (FA, RD, AD, ADC) connectome, the connectome matrix was weighted by the respective metrics through a multi-step procedure. For example, for every reconstructed streamline, the average FA value was computed by sampling the underlying FA image along its trajectory. Then, as each streamline was assigned to connectome edges (connections between predefined brain regions), its contribution to that edge's weight was modulated (multiplied) by its calculated mean FA value. Finally, the strength of each edge in the connectome matrix was determined by averaging the FA-weighted contributions of all streamlines connecting the respective pair of regions. Overall, this process resulted in a total of five different types of connectivity metrics (connectomes) per participant, reflecting different aspects of the white matter structural integrity (O'Donnell & Westin, 2011). For example, FA quantifies the directionality of water diffusion, with higher values indicating greater structural organization. AD – which corresponds to the principal eigenvalue of the diffusion tensor (λ_1) – measures diffusion along the primary direction of the axonal tracts, reflecting axonal integrity. RD measures diffusion perpendicular to the principal eigenvalue, sensitive to myeline integrity $(\lambda_1 + \lambda_2)/2$. ADC, otherwise known as mean diffusivity, represents the overall magnitude of water diffusion within the voxel $((\lambda_1 + \lambda_2 + \lambda_3)/3)$, indicative of general tissue characteristics.”

9. The authors evaluate the likelihood of obtaining results of $p < .05$ across replication and discovery. This is certainly one definition of replicability, but another (which perhaps is even more relevant) is how much the effect itself replicates across datasets. The authors should also compare the magnitude of effect size across sample sizes to elucidate this key property.

We thank the reviewer for this comment. We now also report the average effect sizes (as Pearson's correlation) across the replicable sample sizes. We have also added new information with suggestion from the reviewer namely, **figure 1E, 1G, supplementary tables 5-9 (for all effect sizes)**. These findings are also incorporated in the **results** section (**line 119-128**). An example is shown below:

E) SC – Effect sizes

E) Effect sizes in the SC models. The plot shows the relationship between average effect sizes (i.e. average between effect sizes achieved in the discovery and replication datasets) versus delta (i.e. discovery effect sizes – replication effect sizes). The plot demonstrates that as the mean effect sizes increases, the delta decreases, especially for the phenotypes that were replicable under 425 samples (blue dots). For the rest, it can also be observed that for lower sample sizes, the delta is larger (for average $r < 0.1$).

Minor points:

10. Third paragraph of Main text: R2 values are listed as percentages (e.g. 51.7%) when they should be provided as decimals (e.g. .517).

We have now corrected this.

11. Figure caption: typo “guven””given”.

We have now corrected the spellings.

Reviewer #3 (Remarks to the Author):

Brief summary of the manuscript

In this manuscript, Kotikalapudi et al. evaluate the replicability of using diffusion-weighted imaging (DWI) data to predict trait-like phenotypes. The DWI measures analyzed include structural connectivity (SC), fractional anisotropy (FA), radial diffusivity (RD), axial diffusivity (AD), and apparent diffusion coefficient (ADC). The study examines 58 behavioral and psychometric phenotypes derived from the Human Connectome Project. Through 100 permutations with varying participant sample sizes, the authors estimated the effect size using the coefficient of determination R^2 and employed Killeen's formula as a replicability metric. They concluded that using DWI data for prediction is challenging for traits related to emotions and personality but feasible with a sample size of a few hundred participants for phenotypes related to cognition, motor skills, and demographics.

Overall impression of the work

This study is contextualized within the growing body of research on Brain-Wide Association Studies (BWAS), which has been argued to require sample sizes in the thousands, as suggested by previous work such as Marek et al. (2022). The authors use established methods—such as train-test splitting, regularization, permutation, and Killeen's probability to assess replicability—and provide their Python code openly on GitHub.

The technical rigor of the study could be improved. While the authors' optimistic conclusion regarding sample size requirements for some phenotypic traits is noteworthy, it relies heavily on the robustness of their methodology, which requires closer scrutiny. This raises concerns about the strength and validity of their conclusions. Furthermore, more detailed descriptions of the models used would improve transparency and help prevent potential misinterpretations.

Thank you for acknowledging the relation of our work to the literature and our use of established methods.

In the revised manuscript, we have improved the technical rigor in multiple ways. First and foremost, we have reproduced parts of our results in an independent dataset (**Figure 1G, see also R2C5**). Furthermore, we have investigated the effect of analysis decisions, like the choice of brain atlas or machine learning model (please see replies to R2C6, R2C7), all suggesting that our estimates are generalizable and maybe even slightly conservative, with regards to improved methodology. Finally, all the applied methods are now described in detail in the **methods** section by adding **replicability assessment machine learning pipeline** and **effect size-based theoretical sample sizes** and also expanding the section on **DWI image processing** (please see R2C8). We apologize for the lack of detail in the previous version, which was meant to adhere to a “short communication” publication format, with restricted word count.

Specific comments

A. Effect Size and Statistical Significance

1) The replicability in this study is measured using Killeen's replication probability, which estimates the likelihood that a significant effect size observed in the training dataset will be replicated in the testing dataset. However, Killeen's replication probability is based on a significance threshold of 0.05, an arbitrary cut-off that can lead to an artificial dichotomy between significant and non-significant results.

We appreciate the reviewer's insightful comment on Killeen's p_{rep} and its reliance on the conventional $p < 0.05$ significance threshold. While we acknowledge the inherent arbitrariness of any fixed cutoff, Killeen's p_{rep} was specifically employed to offer a direct probabilistic estimate of whether an initially observed 'significant' finding as defined by this widely adopted standard would replicate its significance. We recognize, theoretically, that the choice of this initial alpha threshold influences the landscape of findings to which p_{rep} is applied; a more stringent alpha (e.g., $p < .01$) would mean p_{rep} is calculated for effects with stronger initial evidence, likely yielding higher p_{rep} values on average, whereas a more lenient alpha would include weaker initial effects, correspondingly associated with lower p_{rep} values. To ensure our interpretation is not solely anchored to this specific framework, we consistently supplement p_{rep} values by reporting and thoroughly analyzing the magnitude of effect sizes (e.g., **supplementary figure 3 (main figure 1E on effect sizes), supplementary tables 5-9**). Our overall conclusions regarding replicability are therefore guided by a comprehensive synthesis of these effect sizes, their behavior across different sample sizes, and their practical relevance, mitigating over-reliance on any single dichotomous outcome. We have now also mentioned this important point in the **method section on Effect size-based theoretical sample sizes** which reads:

“As a caution, it should be noted that while such probabilities are often evaluated against a conventional significance threshold (e.g., $\alpha=0.05$), this specific cut-off can be considered arbitrary and may lead to an artificial dichotomy between significant and non-significant findings.”

2) In the supplemental material, the authors present the formula for the t-statistic of the Pearson correlation coefficient r , which is used to assess statistical significance:

$$t = r * \sqrt{(n - 2) / (1 - r^2)}$$

Here, the adoption of the term $n - 2$, also representing the degrees of freedom for the t-statistic, assumes simple linear regression, where the slope and intercept are estimated. However, in the context of this study, which involves many predictors and regularization, directly applying a t-test, as defined above, may not be appropriate. The complexities introduced by regularization and multiple predictors need to be carefully considered when assessing statistical significance.

We appreciate the reviewer's keen observation regarding the application of the t-statistic formula for Pearson correlation in the context of our multivariate predictive models with regularization. The reviewer is correct in pointing out that the standard t-statistic with $n-2$ degrees of freedom is associated with simple linear regression, estimating a single slope and intercept. It is crucial to clarify that, while the initial model fitting stage involves multiple predictors and regularization techniques, the subsequent assessment of the model's predictive accuracy relies on the univariate relationship between the predicted and observed target values. Even though predicted values in the discovery sample stem from (nested) cross-validation and p-values in the empirical cases are determined by permutation testing, the evaluation of replicability still boils down to testing a simple univariate association. This is coherent with previous work (Marek et al, 2023, Spisak et al., 2024) and justifies the use of the above formula for Killen's replicability. We apologize for the lack of clarity on this, we have now clarified this in the **method** section:

“Effect size-based theoretical sample sizes

To provide guidance on the sample size required for achieving a high probability of successful replication (P_{rep}), we estimated theoretical sample size needs. This estimation was based on the observed effect sizes in our replication samples and primarily assesses the univariate association between predicted outcome values and actual observed target values. This fundamental relationship is maintained even when predicted values are derived from complex methodologies like nested cross-validation and empirical p-values are determined by permutation testing.”

3) The definition of replicability used here focuses solely on statistical significance (i.e., whether R^2 is significantly different from zero), without sufficient attention to effect size. This could lead to misinterpretation, particularly when small effect sizes are found to be significant. While the authors note in the supplemental material that even a small R^2 (e.g., a few percent) can yield reasonable replicability with a couple hundred participants, the practical implications of such small effects should be carefully evaluated, especially when larger effects are desired for clinical or real-world relevance.

We agree that effect sizes are of key importance in the context of the manuscript. Please kindly also see R2C9, which concerns a related issue. In the revised manuscript, we report effect sizes across the models in detail. In fact, we have added new figures to **Figure 1** for this reason (i.e., **Figure 1E and 1G**, which now depict effect sizes) and have also included **supplementary figure 3**, and **supplementary tables 5-9**. **Furthermore, in our revised discussion section, we now more explicitly evaluate the practical implications of the observed effect size magnitudes, particularly for those phenotypes that replicate but yield smaller r values.** We agree that the overall replicability should be interpreted with caution for small effect sizes, especially for state-like phenotypes (as detailed in supplementary table 4), to ensure transparency and aid accurate interpretation. Regarding the interplay between effect sizes and sample sizes, we agree that interpretations should be cautious for phenotypes that only replicate with very large samples, as these typically indicate smaller effect sizes. Our study highlights the complementary finding: phenotypes that are replicable with modest sample sizes (e.g., 100-200) must, by definition, correspond to larger underlying effect sizes. We are confident that our updated results and the information now provided offer greater clarity on this aspect. Our **discussion** also now reads:

“The strong dependence of replicability on effect size implies that, as substantiated by Gratton et al. (Gratton et al., 2022) and others (Ooi et al., 2024; Spisak et al., 2023), there are “two paths towards reliability”. Next to increasing sample sizes (possibly to the thousands), the reliability of the target phenotype, and its relevance to the brain measure, is also of central importance to improve the replicability. It is therefore crucial to acknowledge that when replicability is achieved primarily through the path of very large sample sizes, this often corresponds to smaller underlying effect sizes attributable to white matter microstructural properties alone. Consequently, the practical significance and real-world utility of such findings necessitate careful evaluation, even if statistical replicability is met. Phenotypes that are not only highly relevant to the brain feature of interest (e.g. trait

measures for DWI) but also display a high test-retest reliability can be expected to display the highest effect sizes and will be replicable with small sample sizes. This has been exemplified by several previously reported multivariate neural signatures that showed strong, replicable effect sizes with various behavioural variables, even though they were trained on small samples, sometimes fewer than 50 participants (Spisak et al., 2020; Wager et al., 2013). We argue that such models are the most promising candidates for individual-level predictions guiding clinical care and personalized medical approaches and our findings hint that the effect sizes and replicability necessary for these applications is not impossible to achieve, at least for the most reliable and stable phenotypes.”

B. Multivariate vs. Univariate Modeling

The manuscript frequently references multivariate modeling. However, it is unclear whether each phenotypic trait is predicted individually using DWI data. If the models predict one trait at a time, the approach would be more accurately described as univariate rather than multivariate. Clarifying this distinction would help avoid confusion. In addition, to enhance understanding and transparency, it should be explained how the regression is handled in cases where binary and discrete traits (e.g., sex or personality traits) are involved.

We appreciate the reviewer's call for clarity regarding our modeling approach. In the sense of the general statistical terminology, we do multiple regression and not multivariate regression. However, in the field of neuroimaging, it is standard terminology to refer to these models as multivariate models. The reason for that is historical: in conventional voxel-wise mass-univariate analyses, the brain data is the dependent variable, to be explained by independent variables that could be phenotypic information or experimentally manipulated variables. The typical application of machine learning includes reversing the direction of inference: phenotypic information becomes the dependent variable, to be explained by multiple, often a very large number of brain features. Thus, while in a strict statistical sense often (but not always) a multiple regression model is fit in such analyses, the neuroimaging community tends to refer to these approaches as a *multivariate* approaches, reflecting the original role of the brain variables in the inference. To avoid any ambiguity, we have explicitly clarified this point in the method section, stating that all replicability analyses were conducted on a per-phenotype basis, utilizing a multivariate model for each prediction. For both binary and discrete traits, we used the regression models as in this context, the discrete scale has a reasonable range and underlying continuous nature. For the binary phenotype, that is ‘sex’, changing to a classifier did not show much difference. So, we simply stick with regression throughout all the variables. This has now been clarified in the methods section. Now the **replicability assessment machine learning pipeline** section from **method’s** part reads the following:

*“For each phenotypic target variable, we employed a multivariate predictive modeling approach. Specifically, we utilized the DWI-derived connectivity measures between multiple brain regions as a set of independent variables to predict a single dependent variable, representing the behavioral or phenotypic measure of interest. Therefore, while the prediction target was a single phenotype at a time, the models themselves leveraged the information contained within the high-dimensional DWI connectome, encompassing connectivity across numerous brain regions. All replicability analyses were conducted on a per-phenotype basis using this multivariate framework. Machine learning models were constructed with linear regression (Ridge/L2 regularization). We employed ridge regression for our multivariate analyses due to its effectiveness in handling the high dimensionality and multicollinearity common in brain connectivity data via L2 regularization, leading to stable and generalizable models, a frequent choice in brain-phenotype studies (e.g. (Cui & Gong, 2018; Dadi et al., 2019; Hoerl & Kennard, 1970; Massett et al., 2023; Spisak et al., 2023)). Compared to computationally demanding non-linear methods requiring extensive tuning, ridge regression offers efficiency and interpretability, aligning with our focus on establishing a robust and replicable baseline with minimal hyperparameter optimization (solely the learning rate). Please see **supplementary table 10** for analyses conducted using non-linear methods such as Support Vector Regression (SVR) and Kernel Ridge.”*

C. Prediction vs. Mechanism

While the primary objective of BWAS is to make predictions based on neuroimaging measures, there is an inherent focus on association, as suggested by the term itself. The authors rightly emphasize that “it is important to consider whether the phenotype of interest is relevant at all to the investigated brain measure.” Although association analyses can identify relationships between variables, they do not imply causation.

To provide deeper insights, it would be valuable for the authors to discuss which specific brain regions or networks are particularly associated with each phenotypic trait (e.g., sex, age, cognition, motor skills). Understanding these associations could help bridge the gap between prediction and mechanistic understanding, potentially guiding future studies that aim to uncover causal links.

While an understanding of the specific features or connections driving these predictions is of great interest, it is established that individual feature weights can be less stable than the overall robust prediction of the target variable (Tian & Zalesky, 2021). Nevertheless, we concur with the reviewer on the importance of exploring the underlying biological mechanisms and key predictors. Therefore, acting on the reviewer's recommendation, we have employed the SHAP (SHapley Additive exPlanations (Marçilio & Eler, 2020)) technique to identify the most influential features contributing to our model predictions. These findings have been integrated into the **results** section called **interpretation of brain-phenotype characterization (supplementary figure 4)**, where we now report the key features for all replicable phenotypes. We trust this additional analysis provides valuable context, even as our study's central focus remains on the crucial aspect of model replicability.

References

- Cui, Z., & Gong, G. (2018). The effect of machine learning regression algorithms and sample size on individualized behavioral prediction with functional connectivity features. *Neuroimage*, *178*, 622-637.
- Dadi, K., Rahim, M., Abraham, A., Chyzyk, D., Milham, M., Thirion, B., Varoquaux, G., & Initiative, A. s. D. N. (2019). Benchmarking functional connectome-based predictive models for resting-state fMRI. *Neuroimage*, *192*, 115-134.
- Geiser, C., Götz, T., Preckel, F., & Freund, P. A. (2017). *States and Traits: Theories, Models, and Assessment* (Vol. 33). Hogrefe Publishing. <https://doi.org/10.1027/1015-5759/a000413>
- Gratton, C., Nelson, S. M., & Gordon, E. M. (2022). Brain-behavior correlations: Two paths toward reliability. *Neuron*, *110*(9), 1446-1449.
- He, T., An, L., Chen, P., Chen, J., Feng, J., Bzdok, D., Holmes, A. J., Eickhoff, S. B., & Yeo, B. T. (2022). Meta-matching as a simple framework to translate phenotypic predictive models from big to small data. *Nature neuroscience*, *25*(6), 795-804.
- Hoerl, A. E., & Kennard, R. W. (1970). Ridge regression: Biased estimation for nonorthogonal problems. *Technometrics*, *12*(1), 55-67.
- Marcilio, W. E., & Eler, D. M. (2020). From explanations to feature selection: assessing SHAP values as feature selection mechanism. 2020 33rd SIBGRAPI conference on Graphics, Patterns and Images (SIBGRAPI),
- Massett, R. J., Maher, A. S., Imms, P. E., Amgalan, A., Chaudhari, N. N., Chowdhury, N. F., Irimia, A., & Initiative, A. s. D. N. (2023). Regional neuroanatomic effects on brain age inferred using magnetic resonance imaging and ridge regression. *The Journals of Gerontology: Series A*, *78*(6), 872-881.
- O'Donnell, L. J., & Westin, C.-F. (2011). An introduction to diffusion tensor image analysis. *Neurosurgery Clinics of North America*, *22*(2), 185.
- Ooi, L. Q. R., Orban, C., Nichols, T. E., Zhang, S., Tan, T. W. K., Kong, R., Marek, S., Dosenbach, N. U., Laumann, T., & Gordon, E. M. (2024). MRI economics: Balancing sample size and scan duration in brain wide association studies. *bioRxiv*.
- Pervaiz, U., Vidaurre, D., Woolrich, M. W., & Smith, S. M. (2020). Optimising network modelling methods for fMRI. *Neuroimage*, *211*, 116604.
- Sharot, T., Rollwage, M., Sunstein, C. R., & Fleming, S. M. (2023). Why and when beliefs change. *Perspectives on Psychological Science*, *18*(1), 142-151.
- Snoek, L., van der Miesen, M. M., Beemsterboer, T., Van Der Leij, A., Eigenhuis, A., & Steven Scholte, H. (2021). The Amsterdam Open MRI Collection, a set of multimodal MRI datasets for individual difference analyses. *Scientific data*, *8*(1), 85.
- Spisak, T., Bingel, U., & Wager, T. D. (2023). Multivariate BWAS can be replicable with moderate sample sizes. *Nature*, *615*(7951), E4-E7.
- Spisak, T., Kincses, B., Schlitt, F., Zunhammer, M., Schmidt-Wilcke, T., Kincses, Z. T., & Bingel, U. (2020). Pain-free resting-state functional brain connectivity predicts individual pain sensitivity. *Nature communications*, *11*(1), 187.
- Thornton, M. A., Weaverdyck, M. E., Mildner, J. N., & Tamir, D. I. (2019). People represent their own mental states more distinctly than those of others. *Nature communications*, *10*(1), 2117.
- Thornton, M. A., Weaverdyck, M. E., & Tamir, D. I. (2019). The brain represents people as the mental states they habitually experience. *Nature communications*, *10*(1), 2291.
- Tian, Y., & Zalesky, A. (2021). Machine learning prediction of cognition from functional connectivity: Are feature weights reliable? *Neuroimage*, *245*, 118648.
- Wager, T. D., Atlas, L. Y., Lindquist, M. A., Roy, M., Woo, C.-W., & Kross, E. (2013). An fMRI-based neurologic signature of physical pain. *New England Journal of Medicine*, *368*(15), 1388-1397.
- Woo, C.-W., Chang, L. J., Lindquist, M. A., & Wager, T. D. (2017). Building better biomarkers: brain models in translational neuroimaging. *Nature neuroscience*, *20*(3), 365-377.

On the replicability of diffusion weighted MRI-based brain-behavior models – 2nd Response to reviewers

Dear Editor-in-chief and respected reviewers,

Thank you very much for your valuable consideration and the positive feedback on our revised manuscript. Based on your thoughtful comments, we have now extended our study with an analysis of confounder bias in the HCP dataset and a replicability evaluation of FA, RD, AD and ADC-based measures in the second sample. While the new results do not change our general conclusions about the overall replicability of DWI-based models, they highlight that following best data acquisition standards (as in the HCP dataset) and appropriate handling of confounders are essential next steps towards building DWI-based predictive models with true translational potential. Please kindly find our point-by-point response to reviewer comments below. We are looking forward to hearing from you.

Sincerely,

Raviteja Kotikalapudi and Tamas Spisak, on behalf of all the authors

Reviewer #1

1. Did the authors control for intracranial volume in any way? I understand the prediction (as is the focus here) is different from trying to draw an inference about the SC-phenotype association. However, I could see intracranial volume, which varies as a function of known confounds/colliders (e.g., SES), acting as a hidden confound in the prediction of these phenotypes from structural data.

Response – As the reviewer rightly points out, the primary focus of this study lies in predictive strengths, not inferential analysis of SC-phenotype associations. However, we agree that confounder biases, such as intracranial volume, are critical to consider in predictive modeling. To acknowledge this comment, we have now performed a dedicated confounder analysis on SC in the HCP dataset, using the Spisak (2022) toolbox called “mlconfound” (statistical quantification of confounding bias in machine learning models). The outcome revealed that while most replicable phenotypes were partially confounded by intracranial volume (*revealed by the partial confounder test*), none of the predictions – more importantly – were not entirely driven by this confounder (*revealed by the full confounder test*). This reinforces the value of structural connectivity features for brain-behaviour characterizations. Please see the new **supplementary information 1. Confounder testing for the predictive models**. We now discuss these results – and the importance of testing and handling intracranial volume as a confounder in DWI-based predictive models as follows:

In the methods:

“Additional confounder testing of predictive models

While the primary focus of this work is on the predictive yield and replicability strengths of diffusion-weighted imaging-based connectome models, it is crucial to consider that several phenotypes, and consequently their predictions, can be confounded by total intracranial volume (TIV). To address this potential bias, we performed a confounder analysis using “mlconfound”¹¹. This analysis involved two key aspects, a) Partial confounder test – Assesses if predictions are partially driven by the confounder. A $p < 0.05$ indicates a significant partial influence of the confounder. B) Full confounder test – Determines if predictions are entirely driven by the confounder. A $p < 0.05$ in this test suggests that the predictions are not solely explained by the confounder, implying that the features of interest (e.g., structural connectivities) still hold significant predictive strengths. For this analysis, three variables were utilized: (1) y , the actual behavioral target; (2) \hat{y} , the structural connectivity-predicted behavioral outcome; and (3) c , the total intracranial volume. To obtain y hat, the predictive model was trained and tested on the entire sample size of human connectome project.”

In the results:

“Additional confounder testing on the human connectome project

*While the current study’s primary focus is replicability, there are several other important requirements that a DWI-based biomarker candidate must fulfil. In an additional analysis, detailed in **supplementary tables 16 and 17**, we performed an initial evaluation of one of these important requirements, confounding bias. Specifically, we investigated how much our predictions within the Human Connectome Project dataset were biased by total intracranial volume (TIV) with a novel statistical test for detecting confounding bias in multivariate predictive models (Python package “mlconfound”)¹¹. While our results indicated that none of the phenotypes were fully driven by TIV, most of them were found to be partially biased towards TIV to some degree. While these results are initial in their nature, they highlight that the appropriate handling of confounders – including, but not limited to, TIV – are an essential next step towards building DWI-based predictive models with true translational potential.*

In the discussion:

“While streamline count (SC), or streamline-based connectivity, consistently yielded more replicable phenotypes compared to other connectome metrics (FA, RD, AD, and ADC) across both datasets, we acknowledge that these findings might be specific to the processing pipeline used in this work. Since SC is heavily dependent on analysis parameters such as the tractography algorithm, the number of seeds, and the total number of streamlines, our results should be interpreted with caution. Furthermore, we must consider that the sample sizes required for replication could also be influenced by the presence of confounders. Therefore, future studies should not only continue to explore the predictive power of other diffusion metrics – as they are crucial for a comprehensive understanding of brain mechanisms – but also investigate the specific effects of these confounders on the required sample sizes for replicable findings.”

2. I like the addition of Figure 1C. It is very informative and a nice way to display a lot of data. What strikes me is the pattern of putative replicability in more moderate samples. Age, gender, cognition are the apparent winners. However, as mentioned in the previous point, how much of this could be driven by a hidden confound like head size or SES? Said differently, it is not the brain per se predicting gender or cognition, but rather the phenotype of interest varies with the confound and thus the confound is being aliased into the brain prediction.

Response – We found that the strong association between total intracranial volume (TIV) and gender emerged as a severe confound for gender prediction. Specifically, the predictions explained 94% of variance in gender (y vs. \hat{y}), but also 43% of variance in intracranial volume (\hat{y} vs. TIV). The partial confounder test indicated that the latter is much higher than the expected marginal effect emerging from the correlation between gender and TIV (39%), indicating a highly significant confounder bias ($p < 0.001$). While age and most cognitive measures (as well as many all other phenotypes) also showed significant TIV-bias, the amount of TIV bias was small to moderate in case those target variables (y vs. TIV / y vs. \hat{y} ratio smaller than 0.2, except strength which is strongly associated to gender), holding promise for future efforts to establish unconfounded predictive models by using advanced confound-mitigation strategies (e.g. confound regression, see (Kotikalapudi et al., 2023)).

3. Overall, 36% of phenotypes were replicable, but replicability in Figure 1B is defined as replicable at an $n \leq 425$ in at least one SC metric. That seems to be a low bar. A better representation of the data would be the proportion of phenotypes that are replicable in at least one, at least two, three, four, and all 5 SC metrics. The authors somewhat get at this in Supplementary table 3. However, it's not clear if, for example, the 16 phenotypes that are replicable with ADC are completely overlapping with RD. My guess, based on Supplementary Figure 2, is that there is a lot of overlap, but it would be good to detail. Likewise, for the heatmap in Figure 1C, is this showing the probability of replication for one of the SC metrics? Or just the best performing for each phenotype?

Response – Our intention with Figure 1B was to provide an **overall view of phenotype replicability across all structural connectivity (SC) metrics**, demonstrating the general capacity of DWI for replication within our study. We agree, however, that an additional unfolded representation would enhance clarity and better convey the robustness of our findings.

To address this, we have now provided **two additional supplementary tables (Supplementary Table 4-5)**. These tables offer a detailed yet compact representation of the data, specifically designed to:

1. **Unfold the replicability overview presented in Figure 1B**, detailing the proportion of phenotypes replicable across one, two, three, four, and all five DWI-based connectome metrics.
2. **Clarify the overlap (or non-overlap) of replicable phenotypes** across different metrics (e.g., whether phenotypes replicable with ADC are also replicable with RD), thereby expanding upon Supplementary figure 2/table 3.

We believe these additions provide a more comprehensive understanding for the readership, thoroughly detailing the replicability landscape across individual and combined DWI metrics. Figure 1C shows the probability of replication for the streamline-based connectivity approach – which produced more number of replicable phenotypes than the rest of the DWI-based derived metrics. We have now corrected the header of 1C to avoid any confusion.

4. The authors should make it clear in the text that when an $n < 425$ is referenced that is assuming an n of 425 in both the train and test set of data. As such, these estimates are based on a total sample size of 850, for example. However, it should not be assumed that a model will replicate/generalize equally well to a brand-new sample of smaller N (e.g., 50). The language used around this topic throughout the abstract and main text is ambiguous. The authors should either make it clear that the sample refers to the training sample size or (more preferably) make it crystal clear that their estimates are based off a total sample size twice as big as what they are listing. This is especially important as quality reference samples used to benchmark replicability is dependent on its size.

Response – We understand the reviewer's concern. We have now modified the abstract and explicitly mentioned that sample size refers to the discovery samples and total sample size = discovery + replication samples. We have also added and highlighted this knowledge, most importantly in the Method's section of replicability assessment machine learning pipeline. We have also explicitly mentioned this in the very first line of the Discussion to set the tone with more clarity. We have now explicitly mentioned it in the caption of Figure 1A as well.

Reviewer #2

The authors have greatly improved the manuscript with this revision by expanding their analyses, providing additional effect size information, and adding an additional dataset. The new Table 1 is itself a major improvement to the description and clarity of the findings. That said, I do have a couple of lingering concerns, which I consider to be minor.

Response – We thank the reviewer for the valuable feedback. We have now addressed the important concerns raised here.

1. The primary metric highlighted, and the only one used in the second AOMIC dataset, is streamline count (SC). My concern is that SC is arguably the least quantitative of the available measures, in the sense that it is the least likely to be numerically stable across different tractography processing decisions. The number of streamlines may depend heavily on the seeding scheme (including the random seed used, the number of seeds placed, etc.), as well as on whether tractography is constrained by the number of placed seeds or the total number of streamlines derived. It is possible that the authors would recommend that future studies use the exact tractography procedures employed here, but that feels overly narrow. I do understand that SC performed best in the presented analyses, but given its potential limitations, this warrants more discussion. It would also be helpful to see how other metrics, such as FA, generalize to the new dataset. Regardless of the outcome, this would be informative (if other measures also generalize well, that supports broader applicability; if not, it would help clarify the specificity of the findings).

Response – Thanks for your valuable feedback on our choice of streamline count (SC) and your concerns about its stability. While our initial focus on SC was for hypothesis-based reductionism, we agree that other metrics like FA, RD, AD, and ADC hold significant biological relevance. To address this, we have now computed and evaluated all these metrics, re-checking sample size requirements for replicability. As detailed in **Supplementary Table 15**, only sex and BMI phenotypes were consistently replicable across these additional measures. We have expanded our discussion (second last paragraph) in the revised manuscript to provide a more comprehensive context on these findings and their implications for future research.

2. In Figure 1, panels D, E, and F appear to indicate that trait and state measures will be represented in black vs. purple based on the text colors, but that is not reflected in the actual figure panels. Additionally, in panel C, the blue color scale represents P-replicability, whereas in panel E it represents sample size. It is unclear what blue represents in panels D and F. Overall, the use of color in this figure could be improved to make this sole main text figure more intuitive and easier to interpret.

Response – We thank the reviewer for the suggestions. The colors are now matched and in sync with the legends as well. In order to avoid confusion, we have simplified the figure C, by choosing standard grey scale. Also, replicable phenotypes are always in greens and non replicable in reds (e.g., B, E, G-1, G-3). In general, blue is for state-like phenotypes (D, F). Figure is presented here for a quick reference.

Reviewer #3

The authors have responded thoroughly and thoughtfully to the reviewers' comments. I am satisfied with the revisions and commend the authors for their careful attention to detail. I have just one minor grammatical suggestion. In the following sentence, the first "the" appears to be a typographical error and should likely be replaced with "of": "Our primary replicability analyses on the HCP dataset utilized the 84-region Desikan-Killiany brain atlas, which is the default option the the software FreeSurfer." The corrected sentence should read: "...which is the default option of the software FreeSurfer."

Overall, I find the revised manuscript suitable for publication, subject to this minor correction and the final evaluation by the editors and other reviewers.

Response – We sincerely thank the reviewer for the valuable first round of reviews and their final evaluation. The typo is corrected.

Kotikalapudi, R., Kincses, B., Zunhammer, M., Schlitt, F., Asan, L., Schmidt-Wilke, T., Kincses, Z. T., Bingel, U., & Spisak, T. (2023). Brain morphology predicts individual sensitivity to pain: a multicenter machine learning approach. *Pain*, 164(11), 2516-2527.